# IL-15 and PIM kinases direct the metabolic programming of intestinal intraepithelial lymphocytes

Olivia J. James[1], Maud Vandereyken[1], Julia M. Marchingo [2], Francois Singh[1], Susan E. Bray[3], Jamie Wilson[4], Andrew G. Love[1] & Mahima Swamy [1✉]

Intestinal intraepithelial lymphocytes (IEL) are an abundant population of tissue-resident T cells that protect and maintain the intestinal barrier. IEL respond to epithelial cell-derived IL-15, which is complexed to the IL-15 receptor α chain (IL-15/Rα). IL-15 is essential both for maintaining IEL homeostasis and inducing IEL responses to epithelial stress, which has been associated with Coeliac disease. Here, we apply quantitative mass spectrometry to IL-15/Rα-stimulated IEL to investigate how IL-15 directly regulates inflammatory functions of IEL. IL-15/Rα drives IEL activation through cell cycle regulation, upregulation of metabolic machinery and expression of a select repertoire of cell surface receptors. IL-15/Rα selectively upregulates the Ser/Thr kinases PIM1 and PIM2, which are essential for IEL to proliferate, grow and upregulate granzyme B in response to inflammatory IL-15. Notably, IEL from patients with Coeliac disease have high PIM expression. Together, these data indicate PIM kinases as important effectors of IEL responses to inflammatory IL-15.

[1] MRC Protein Phosphorylation and Ubiquitylation Unit, University of Dundee, Dundee, UK. [2] Division of Cell Signalling and Immunology, School of Life Sciences, University of Dundee, Dundee, UK. [3] NHS Research Scotland, Tayside Tissue Biorepository, University of Dundee, Dundee, UK. [4] Department of Pathology, NHS Tayside, Ninewells Hospital, Dundee, UK. ✉email: m.swamy@dundee.ac.uk

ntraepithelial lymphocytes (IEL) are a specialised lymphoid compartment in the intestinal epithelium, comprising a heterogeneous population of T lymphocytes[1]. IEL express either the αβ T cell antigen receptor (TCR) or the γδ TCR, alongside TCR co-receptors CD8αβ or CD8αα and to a lesser extent CD4 $^{(+/-)}$. The most prevalent IEL subsets within the epithelium of the mouse small intestine are those expressing TCRγδ and CD8αα (TCRγδCD8αα), which account for ~50% of the total IEL pool, with the remaining mostly being TCRαβCD8αβ or TCRαβCD8αα-expressing cells. In human guts, TCRαβCD8αβ and TCRαβCD4 make-up the majority of IEL, with TCRγδ IEL contributing <5–20% of the IEL compartment[2]. IEL are constantly exposed to commensal bacteria, dietary antigens, and potential pathogens at the intestinal epithelium, and are tasked with responding to epithelial stress and protecting the intestinal barrier from external insults. How IEL get activated is a matter of debate; but IEL do not solely depend on TCR stimulation for their activation, rather, signals from the microenvironment are important for communicating compromised barrier integrity to the surrounding IEL and eliciting an effective immune response. One such signal is the common γ-chain (γc) family cytokine IL-15.

IL-15 is produced by a wide range of cells including non-hematopoietic cells such as intestinal epithelial cells (IEC) and its expression is elevated in the gut microenvironment during tissue stress or infection[3]. IL-15 is commonly presented to surrounding IEL in a cell contact-dependent manner, known as trans-presentation, by IEC expressing IL-15 bound to the high-affinity IL-15 receptor α subunit (IL-15Rα)[4]. IL-15/Rα interacts with the β (CD122) and γc subunits present on the surface of IEL, initiating JAK/STAT-mediated signalling events that alter lymphocyte function[5]. It is well established that IEL require IL-15 for their survival in the small intestine[6], and development and maturation of CD8αα IEL can be rescued in IL-15Rα$^{-/-}$ mice by restoring IL-15Rα expression to the intestinal epithelium[4].

IL-15 impacts more than just IEL survival. IL-15 induces proliferation and cytolytic effector function in human IEL in vitro[7]. Importantly, elevated intestinal IL-15 expression is associated with increased numbers of cytotoxic IEL in the small intestine of Coeliac disease (CeD) patients[8], attributed to IL-15-driven survival and proliferative expansion of IEL[9,10]. CeD is an autoimmune enteropathy whereby genetically susceptible individuals have an adverse reaction to gluten, causing immune-mediated damage to the small intestine. IEL derived from patients with CeD have elevated expression of activating NK cell receptors such as NKG2D and CD94 and expand massively in response to elevated levels of IL-15[9,11]. These findings suggest that regulation of epithelial expression of IL-15 is a key mechanism by which IEL activity is controlled. Thus, for IEL to utilise IL-15 as a survival stimulus but also respond to rising levels of IL-15 as a 'danger signal', IEL responses to IL-15 must be tightly controlled. However, the mechanism by which IL-15 mediates IEL function is poorly understood.

Much of the data we have on effects of IL-15 on IEL have been derived from CeD patients and from 'healthy' IEL obtained from oesophagus-gastro-duodenoscopies for non-CeD complaints, or from gastric bypasses for morbidly obese patients[11–13]. Due to the cell contact-dependent mechanism of IL-15 presentation, it is likely that IEL from CeD patients, and indeed, other entero-pathies, receive additional stress signals from the IEC that they are in contact with in vivo. For example, the stress-induced NKG2D ligands, MIC-A and MIC-B, are elevated on damaged IEC in CeD, and can activate IEL[11,12,14]. IEL are also exposed to IL-21 and interferons in CeD[15,16], and it is unclear how many of these factors are present in so-called healthy human controls. Hence, it is not currently clear how IL-15 directly affects IEL in the absence of any other stress-related stimuli.

Here, we investigate the intrinsic effects of exposure to elevated levels of IL-15 on IEL. We take a systematic unbiased approach using high-resolution quantitative proteomics to define the global changes induced in purified TCRγδCD8αα, TCRαβCD8αα and TCRαβCD8αβ IEL subpopulations after 24 h exposure to IL-15. Our data show how IL-15 regulates the activation status of IEL through the upregulation of various activating and inhibitory receptors, biosynthetic and bioenergetic activation, and induction of proliferation. Importantly, these data reveal a critical role for the proto-oncogenes PIM1 and PIM2 kinases in IL-15-induced proliferation, growth, and acquisition of effector function in IEL.

## Results

**Proteome profiling reveals distinct features of IL-15/Rα stimulated IEL.** Here, we used quantitative label-free high-resolution mass spectrometry to explore how IL-15 shapes the proteome of the main mouse IEL subsets: TCRαβCD8αβ, TCRαβCD8αα and TCRγδCD8αα (Supplementary Fig. 1a). Importantly, we used IL-15/Rα complexes to better mimic the trans-presentation of IL-15 by IEC, and because it is more representative of physiological conditions[17,18]. For IL-15-induced activation, IEL were treated for 24 h with 100 ng/mL IL-15/Rα (3.4 nM), based on pSTAT5 induction (Supplementary Fig. 1b) and higher survival (Fig. 1a) observed at this concentration. Untreated controls were derived directly ex vivo, as IEL do not survive well in culture with low concentrations, or in complete absence of IL-15 (Fig. 1a).

We identified and quantified >7500 proteins in the total dataset (4 biological replicates/population; ±IL-15/Rα), providing high-resolution quantitative proteomes of three developmentally diverse IEL populations. Intensities were converted into estimated protein copy number per cell using the proteomic histone ruler method[19,20]. Both intensities and copy numbers showed good correlation between replicates and subsets, despite their differences in ontogeny. All IEL subsets had similar protein content (~40–50 pg/cell), and this was not significantly changed by IL-15 (Fig. 1b). These data indicate that different IEL subsets are similar in their protein make-up and overall protein abundance (Supplementary Fig. 1c, d).

To investigate how IL-15 differentially altered the proteomic landscape of each subset, we calculated the fold change in expression of proteins (IL-15-treated vs untreated) and found that IL-15 largely drove the upregulation of proteins in all IEL subsets (Fig. 1c and Supplementary Data 3). Interestingly, IL-15 had the strongest impact on TCRαβCD8αβ IEL, with >600 proteins being significantly upregulated >2-fold, approximately twice as many as in TCRαβCD8αα and TCRγδCD8αα IEL. This was surprising, as TCRαβCD8αβ IEL expressed the lowest copies of IL2Rβ (CD122), the IL-15 receptor subunit necessary for signal transduction downstream of IL-15 (Supplementary Fig. 1e). However, the phosphorylation of STAT5 both ex vivo and in response to IL-15/Rα stimulation was comparable in all three subsets (Supplementary Fig. 1f). These data suggest that CD122 levels on TCRαβCD8αβ IEL are sufficient to activate downstream signalling at a similar level to that seen in the other IEL subsets.

To identify shared IL-15 targets in IEL, we next asked which proteins were commonly regulated by IL-15 among IEL subsets. IL-15/Rα stimulation led to the downregulation of relatively few proteins, with only 5 being significantly downregulated >2-fold in all three IEL subsets (Fig. 1d, Supplementary Data 1). On the other hand, we identified 95 proteins that were significantly upregulated >2-fold in all IEL subsets, and >240 proteins significantly upregulated in 2 or more subsets (Fig. 1e). Functional annotation clustering revealed that the IL-15-upregulated proteome was most enriched for proteins involved in processes

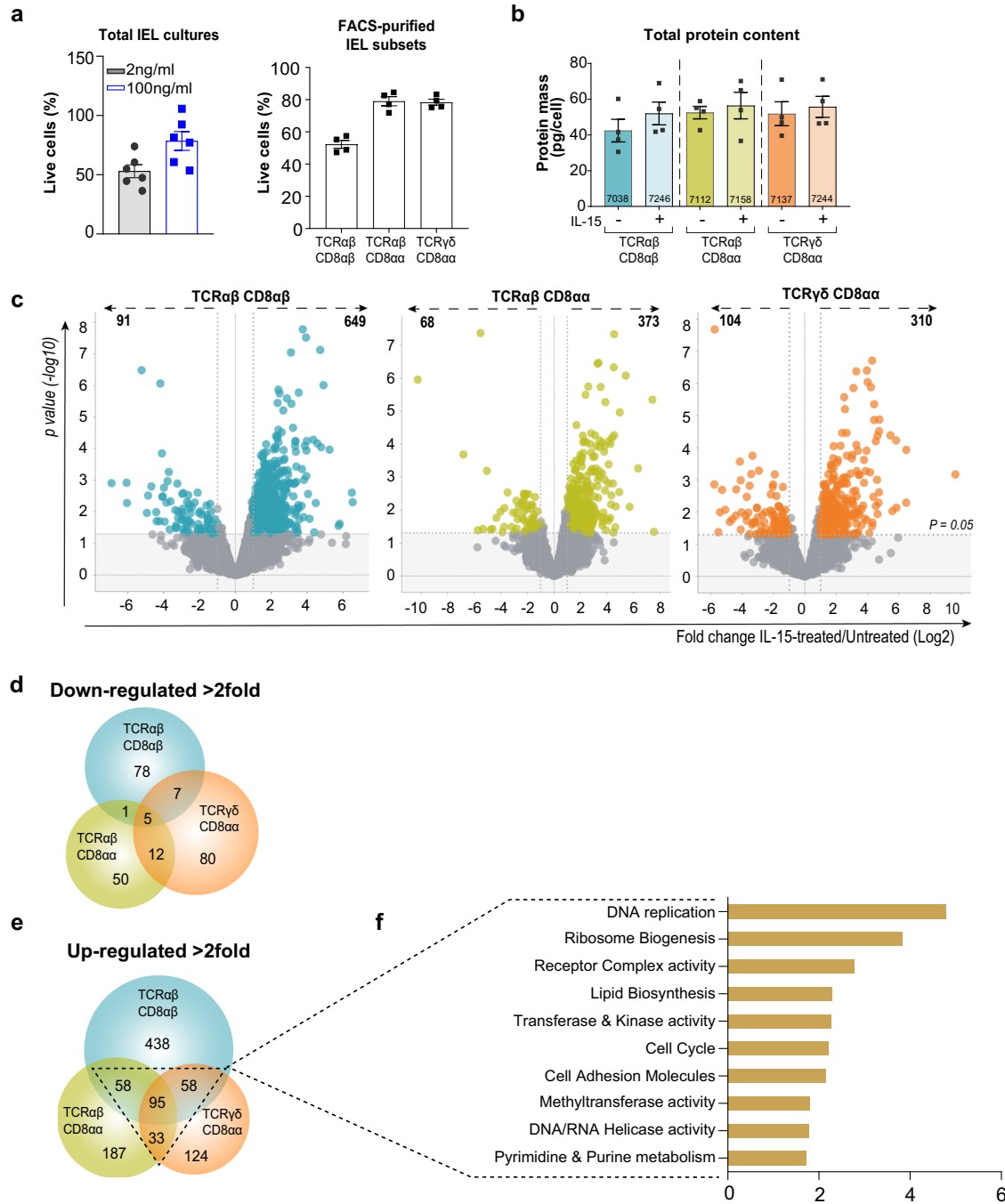

**Fig. 1 Global effects of IL-15/Rα stimulation on the proteomes of the three main IEL subsets. a** Bar chart shows the percentage of live cells following 24 h IL-15/Rα (100 ng/mL) stimulation of CD8[+] IEL (left, $n = 6$ biologically independent experiments) and Fluorescence activated cell sorting (FACS)-purified IEL subsets (right, $n = 4$ biologically independent samples) (gating strategy shown in Supplementary Fig. 1a). Percentages were calculated from the number of cells that were considered live (negative for DAPI staining) following IL-15/Rα treatment for 24 h, divided by the number of cells seeded for culture (1 million/mL). **b** The protein content (pg/cell) of each IEL subset ± 24h IL-15/Rα stimulation. The number of proteins identified by average copy number expression in each subset are displayed on the respective bars ($n = 4$ biologically independent samples). **c** Volcano plots show the differential expression of proteins following IL-15 stimulation for each IEL subset. Data are presented as the distribution of the copy number ratio (IL-15-treated vs untreated) (log2 (fold change)) against the inverse significance value ($-$log10($p$-value)). Proteins were considered differentially expressed following IL-15 stimulation if the log2 fold-change value was >1 (>2-fold) in either an upregulated or downregulated direction. The grey area depicts the cut-off for proteins deemed to have an insignificant fold change ($p$-value >0.05). Hence, proteins in colour were deemed significantly changed following IL-15 stimulation. Statistical significance was derived from two-tailed empirical Bayes moderated $t$-statistics performed in limma on total proteome, see Supplementary Data 3. **d, e** Venn diagrams show the commonality of proteins significantly downregulated and upregulated (>2-fold), respectively, between all IEL subsets following 24 h IL-15/Rα stimulation. **f** Top 10 functional clusters enriched in proteins that were commonly 2-fold upregulated by IL-15/Rα stimulation in at least 2 IEL subsets, as in **e**. See also supplementary Data 2. All error bars are mean ± s.e.m.

such as DNA replication and repair; cell cycle; and ribosome biogenesis (Fig. 1f and Supplementary Data 2). The analyses also highlighted that the upregulated proteins were highly enriched for transferases, including kinases, synthases and methyltransferases; as well as cell surface receptors and adhesion molecules. In summary, these data reveal many previously unrecognised pathways in the IL-15-regulated proteome of IEL.

**IL-15 drives IEL proliferation by triggering G1/S transition.** IEL maintained in 100 ng/mL IL-15/Rα increased in numbers over 96 h (Fig. 2a), whereas low concentrations of the cytokine complex (2 ng/mL) were only enough to maintain a proportion of the starting IEL numbers. This indicated that higher concentrations of IL-15/Rα either enhanced survival or proliferation, or both, in IEL. We found that primary IEL expressed the pro-survival proteins Bcl2, Bcl-XL and Mcl1, and both Bcl2 and Mcl1 were more highly expressed following IL-15/Rα stimulation (Fig. 2b). To assess proliferation, we used CellTrace CFSE and found that IEL started to divide at 48 h in response to 100 ng/mL IL-15/Rα but did not divide at all in low levels, even after 4 days (Fig. 2c). These data support the notion that strong IL-15 signals are required for inducing IEL proliferation, as suggested by the functional annotation clustering (Fig. 1f). We next investigated the expression of cell cycle proteins and found that IL-15-treated IEL had elevated expression of D-type cyclins and the associated cyclin-dependent kinase CDK6 (Fig. 2d, e). To prevent transition into the S phase of the cell cycle, the cyclin-dependent kinase inhibitor 1B (CDKN1B or p27) binds to and inhibits downstream cell cycle targets such as D-type cyclins and CDK6. Our data revealed a downregulation of p27 by IL-15/Rα (Fig. 2e), suggesting that IL-15 drives G1/S transition by downregulating cell cycle inhibitors such as p27 and upregulating positive regulators of the cell cycle. DNA synthesis measurements using EdU in IEL cultured in IL-15/Rα confirmed that only 100 ng/mL triggered entry of IEL into the S phase of the cell cycle after 48 h as compared to 10 ng/mL (Fig. 2f). Overall, these data indicate that strong IL-15/Rα signals trigger proliferation of IEL by regulating key cell cycle checkpoint proteins to allow IEL exit from quiescence.

**IL-15 stimulation triggers biosynthetic pathways in IEL.** A key feature of T cell activation is a massive increase in cell size fuelled by an increase in the uptake of nutrients and metabolic reprogramming[21]. IEL already bear many hallmarks of activation, including high granzyme, CD69 and CD44 expression, but are still small, metabolically quiescent cells[22]. Although we did not detect an increase in the overall protein content at the 24 h timepoint used in the proteomic experiment (Fig. 1b), IEL maintained in 100 ng/mL IL-15/Rα increased in size and granularity by 48 h and continued to grow over 96 h, as compared to those maintained in 2 ng/mL IL-15/Rα (Fig. 3a). Therefore, we interrogated the proteomic data to gain insights into the biosynthetic and bioenergetic machinery and mechanisms by which IL-15 might drive this expansion in cell size.

rRNA synthesis and ribosome biogenesis were highlighted as significantly enriched pathways in the functional annotation clustering (Fig. 1f) and further investigation revealed these proteins largely related to the 90 S pre-ribosome, the earliest stable assembly intermediate in the formation of eukaryotic ribosomes (Fig. 3b). This data corresponded with an overall increase in the total ribosome content (pg/cell) following IL-15 stimulation in all IEL subsets (Fig. 3c). Since increased ribosomal biogenesis should enhance protein translation, we assessed the effects of IL-15 stimulation on protein synthesis using O-propargyl-puromycin (OPP) labelling. OPP is incorporated into nascent polypeptide chains and when labelled with a fluorophore, permits measurement of rates of translation. IEL cultured in 100 ng/mL IL-15/Rα for 48 h showed an increase in protein synthesis after 15 min of OPP-labelling (Fig. 3d). In contrast, cells cultured in 10 ng/mL IL-15/Rα did not show any labelling above cycloheximide (CHX) treated controls. Thus, strong IL-15/Rα signalling triggers increased protein synthesis in IEL by enhancing ribosome biogenesis.

Given the increase in protein synthesis in IL-15-stimulated IEL, we next assessed how IL-15 altered the expression of limiting nutrient transporters that provide biosynthetic precursors for anabolic processes. We observed a significant increase in the expression of various nutrient transporters including the transferrin receptor (TfR/CD71), zinc transporters ZIP6 and ZIP10, large neutral amino acid transporter, SLC7A5/LAT1, and neutral amino acid transporters, SLC38A2/SNAT2 and SLC1A5/ASCT2, upon IL-15/Rα stimulation (Fig. 3e), many of which are key nutrient transporters for T cell activation[23–25]. Using flow cytometry, we confirmed elevated expression of CD71 and for CD98 (SLC3A2), the heavy chain associated with SLC7A transporters, in response to IL-15 stimulation (Fig. 3f). These data suggest that IL-15/Rα enhances the ability of IEL to take up amino acids, the building blocks for protein synthesis, and other nutrients essential for optimal T cell function, such as iron and zinc[26,27].

Conventional T cells increase their capacity for both glycolysis and oxidative phosphorylation (OXPHOS) upon activation to satisfy their increased bioenergetic and biosynthetic demands[21]. IL-15 marginally increased the expression of glucose transporter GLUT1 from <1000 copies to ~2000–3000 copies/cell (Fig. 4a). In contrast to conventional activated CD8 T cells, which preferentially express high levels of GLUT1[28], our proteomic data revealed that IEL have abundant basal expression of the high-affinity glucose transporter GLUT3 (~20,000 copies) (Fig. 4a), matching that expressed on activated CD8 T cells (~25,000 molecules/cell)[28]. We measured glucose levels in the media of IEL cultured with IL-15/Rα and found a decrease specifically in cultures with 100 ng/mL of IL-15/Rα (Fig. 4b). Of the known players in glucose metabolising pathways, we found that IL-15 induced the upregulation of the glucose phosphorylating enzyme hexokinase 2 (HK2)[29] and the lactate exporters MCT1 and MCT3 (Fig. 4c). To investigate whether IEL metabolised glucose through glycolysis, we measured lactate output in IEL cultured in IL-15/Rα and found that 100 ng/mL IL-15/Rα specifically increased production of lactate over 72 h (Fig. 4d). These data indicate that the small, but significant changes in GLUT1, HK2 and lactate transporter expression translate to increased glycolysis in IL-15/Rα-stimulated IEL.

As glucose also contributes to OXPHOS in activated T cells[30], we next assessed mitochondrial respiratory capacity of IL-15-stimulated IEL using high-resolution oxygraphy. The data show that strong IL-15/Rα signals trigger an increase in basal oxygen consumption at the cellular level (Fig. 4e, BASAL). Further analyses were performed in permeabilised cells to permit addition of exogenous mitochondrial complex substrates, inhibitors, and ADP. This approach allowed us to evaluate the contributions of individual mitochondrial complexes to the increased mitochondrial respiration. Based on these analyses, we could attribute the increase in mitochondrial respiration to an increase in maximal OXPHOS respiration through Complexes I + II (GMS$_P$), and to increased electron transfer system capacity (ETC), suggesting an enhanced transfer of electrons from complex II to complexes III and IV. These data indicate that IL-15 stimulation increased mitochondrial function and spare respiratory capacity, which resulted in an overall increase in OXPHOS. Taken together, IL-15 triggered multiple biosynthetic pathways to enable the switch from resting to active IEL.

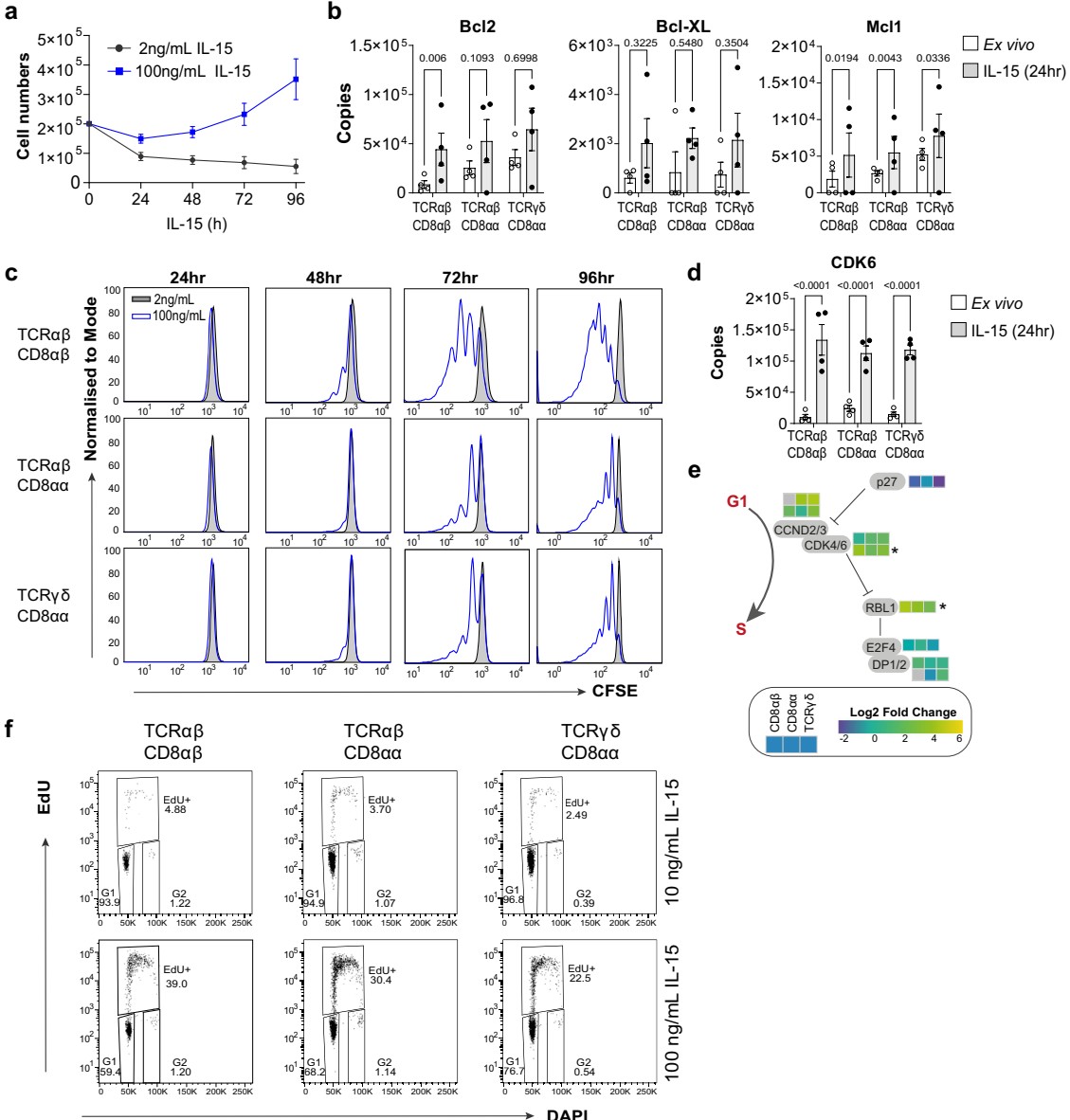

**Fig. 2 IL-15/Rα stimulation drives proliferation of IEL by licensing the G1/S transition. a** Total numbers of live CD8[+] IEL following culture with IL-15/Rα (100 ng/mL or 2 ng/mL) (0–72 h $n = 7$, 96 h $n = 6$ biologically independent experiments). **b** Estimated copy numbers/cell of survival proteins Bcl2, Bcl2l1 (Bcl-xL) and Mcl1 in each IEL subset ± 24 h 100 ng/mL IL-15/Rα stimulation ($n = 4$ biologically independent samples). **c** IEL were stained with CellTrace CFSE prior to stimulation with either 2 ng/mL (grey) or 100 ng/mL (blue) IL-15/Rα for 4 days. Every 24 h cells were stained for subsets; TCRαβCD8αβ, TCRαβCD8αα, TCRγδCD8αα and CFSE expression was analysed by flow cytometry (gating strategy as in Supplementary Fig. 1a). The discrete peaks in the histograms represent successive generations of live, DAPI-negative IEL (representative of $n = 3$ biologically independent experiments). **d** Estimated copy numbers/cell of cyclin-dependent kinase 6(CDK6) ± 24h 100 ng/mL IL-15/Rα stimulation ($n = 4$ biologically independent samples), exact $p$-values are provided in Supplementary Data 3. **e** Heatmap for proteins involved in G1 to S phase transition of the cell cycle. Heatmap squares are corresponding log2 fold-change in copy number expression (IL-15-treated vs unstimulated) for each IEL subset. Asterisks depict significantly changed proteins ($p < 0.05$) identified in the pathway analysis, exact $p$-values can be found in Supplementary Data 2. **f** IEL were stimulated with either 10 ng/mL or 100 ng/mL IL-15/Rα for 48 h. Cells were stained for IEL subsets (gating strategy shown in Supplementary Fig. 6a) and DNA synthesis was assessed by incorporation of Ethynyl-deoxyuridine (EdU) (representative of $n = 2$ biologically independent samples). All error bars are mean ± s.e.m. For all proteomic data (**b, d, e**), statistical significance was derived from two-tailed empirical Bayes moderated $t$-statistics performed in limma on total proteome, see Supplementary Data 3.

**Cytotoxic effector function of IL-15/Rα-stimulated IEL.** Thus far we have focussed on the enriched pathways in the IL-15/Rα regulated IEL proteome. However, a key question in IEL biology is how IL-15 triggers cytotoxicity of IEL towards surrounding IEC to contribute to the pathology of CeD. The enhancement of IEL cytotoxicity by IL-15 has been attributed to an increase in GzmB expression in IEL, and to changes in NK receptor expression. Our proteomic data revealed that all IEL subsets expressed very high

levels of GzmA and GzmB in the resting state (5–20 million and 2–4 million copies/cell, respectively) and levels of perforin that were comparable to effector T cells (ref. [31] and Fig. 5a). IL-15 led to a modest upregulation of GzmB expression, which we confirmed by flow cytometry (Fig. 5b). Interestingly, we observed no difference in GzmA copies but detected a decrease in GzmA by flow cytometry in IEL cultured for 24 h in IL-15/Rα (Fig. 5a, b). We further investigated the expression of several key proteins

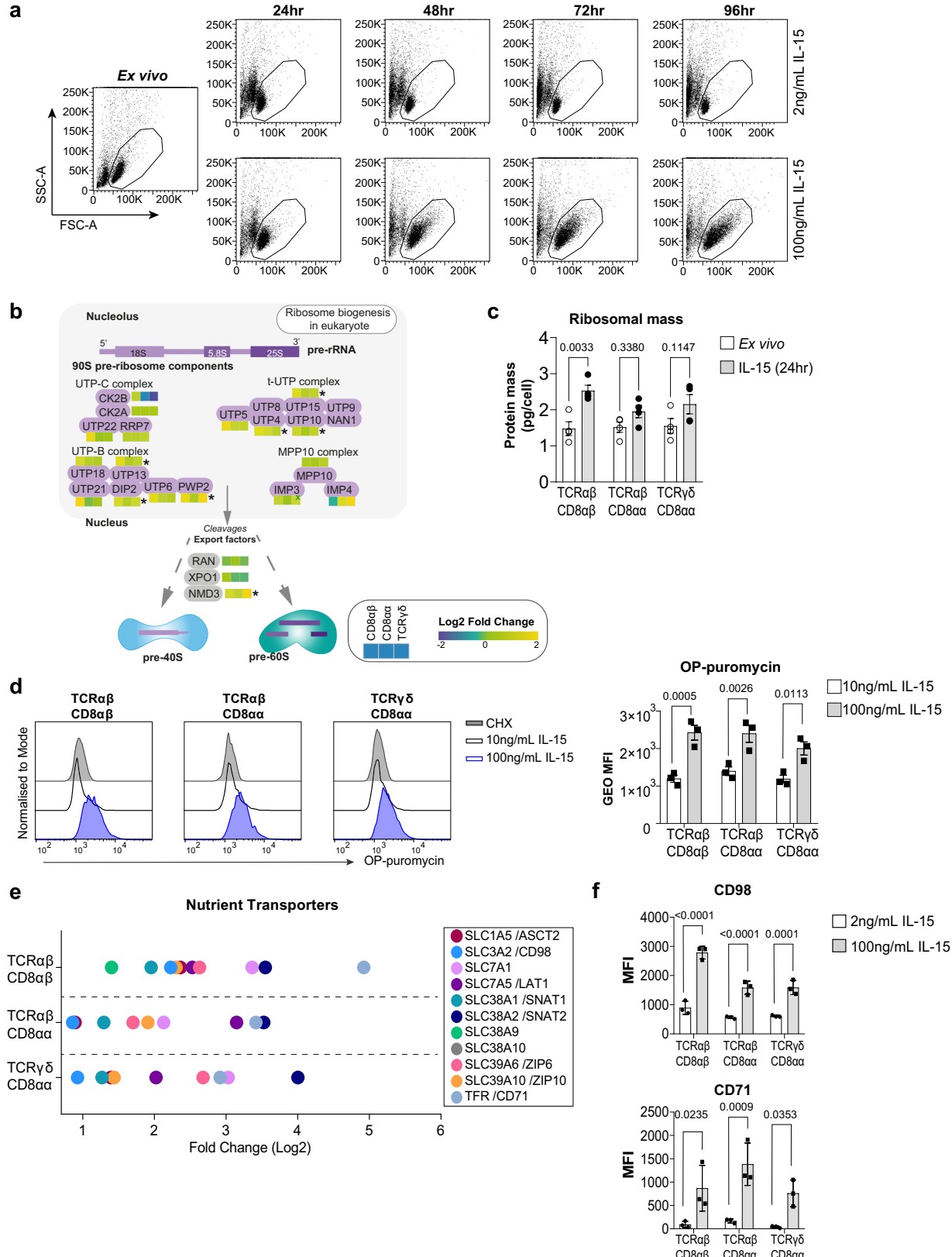

involved in degranulation, including the Munc family of proteins, STXBP2, and Munc13-4 (Unc13d), the Rab GTPase RAB27A and perforin. None of these proteins were regulated by IL-15 (Fig. 5a), confirming the idea that IEL are already fully primed for activation and degranulation.

Given these data, we asked whether IL-15 enhanced cytotoxic activity in IEL, and whether this was dependent on granzymes. Ex vivo IEL only efficiently killed K562 target cells in a redirected killing assay in the presence of 100 ng/mL IL-15/Rα (Fig. 5c). Cytotoxic activity was further increased by additional stimulation

**Fig. 3 IL-15/Rα increases IEL protein synthesis, nutrient uptake and growth. a** Dot plots (representative of $n = 3$ independent experiments) show the forward scatter (FSC, indicator of cell size) vs side scatter (SSC, indicator of granularity) of live IEL over 96 h culture in either 2 ng/mL or 100 ng/mL IL-15/Rα. **b** Heatmap for proteins pertaining to the pre-90S ribosome. Heatmap squares are corresponding log2 fold-change in copy number expression (IL-15-treated vs unstimulated) for each IEL subset. Asterisks depict significantly changed proteins ($p < 0.05$) identified in the pathway analysis, exact $p$-values can be found in Supplementary Data 2. **c** Bar graph shows the protein content (pg/cell) of the sum of ribosomal proteins (GO:0005840) identified in all IEL subsets ± 24 h IL-15/Rα stimulation ($n = 4$ biologically independent samples). Statistical significance was derived from two-way ANOVA with Sidak's multiple comparisons test. **d** OPP incorporation in IEL ($n = 3$ biologically independent experiments) cultured with 10 ng/mL or 100 ng/mL IL-15/Rα for 48 h. As a negative control, incorporation was inhibited by cycloheximide (CHX) pre-treatment in IEL cultured with 100 ng/mL IL-15/Rα. Histograms show OPP incorporation in CHX-treated IEL (grey filled), IEL treated with 10 ng/mL IL-15/Rα (black) and IEL treated with 100 ng/mL IL-15/Rα (blue). Bar graph shows the geometric mean fluorescence intensity (GEO MFI) of OPP in each IEL subset (gating strategy shown in Supplementary Fig. 6b), statistical significance was derived from two-way ANOVA with Sidak's multiple comparisons test. **e** Dot plot shows the log2 fold-change of nutrient transporters significantly ($p < 0.05$) differentially expressed in the proteomic dataset. **f** Flow cytometric analyses of CD98 and CD71 expression from IEL cultured for 72 h in either 2 ng/mL or 100 ng/mL IL-15/Rα (gating strategy as in Supplementary Fig. 1a). Data is presented as mean fluorescence intensity (MFI) ($n = 3$ biologically independent experiments), statistical significance was derived from two-way ANOVA with Sidak's multiple comparisons test. All error bars are mean ± s.e.m. For all proteomic data (**b, e**), statistical significance was derived from two-tailed empirical Bayes moderated $t$-statistics performed in limma on total proteome, see Supplementary Data 3.

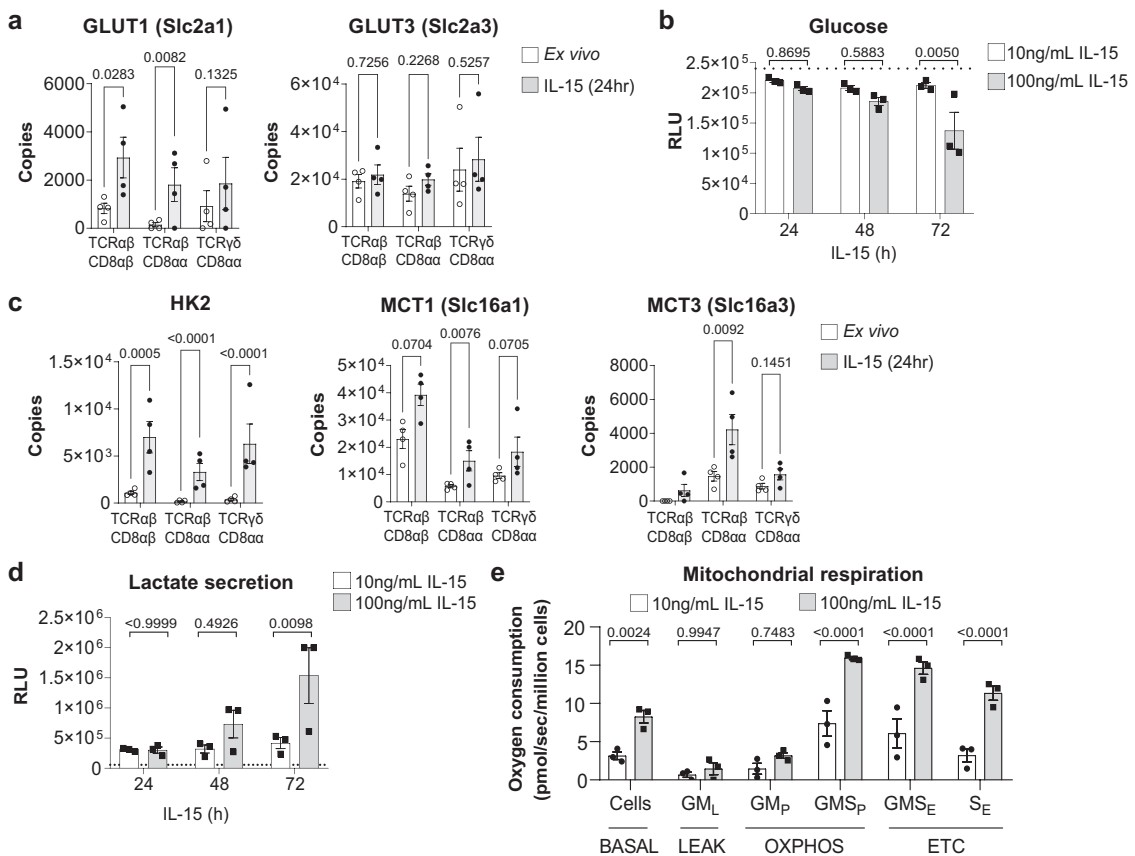

**Fig. 4 IL-15/Rα stimulation increases mitochondrial respiration in IEL. a** Estimated protein copy number/cell of the glucose transporters, GLUT1 and GLUT3, for all IEL subsets ± 24 h IL-15/Rα stimulation ($n = 4$ biologically independent experiments). **b** Bar graph shows the glucose levels remaining in the medium of IEL cultures in the presence of either 10 ng/mL or 100 ng/mL IL-15/Rα for 72 h. Levels of glucose remaining in the medium were assessed by bioluminescence assay, data are presented as relative light units (RLU). The dotted line depicts the signal from the control wells containing only medium but no cells ($n = 3$ biologically independent experiments). **c** Estimated protein copy number/cell of the lactate transporters MCT1 and MCT3 and the glycolytic enzyme hexokinase 2 (HK2). Data is shown for all IEL subsets ± 24 h IL-15/Rα stimulation ($n = 4$ biologically independent samples), exact $p$-values can be found in Supplementary Data 3. **d** Lactate output from IEL cultured as in **b** ($n = 3$ biologically independent experiments). **e** Mitochondrial respiration measurements in IEL cultured in 10 ng/mL100 ng/mL IL-15/Rα for 44 h ($n = 3$ biologically independent samples). Oxygen consumption is expressed as pmol $O_2$ • $s^{-1}$ • million cells$^{-1}$. Respiratory rates were measured in cells (BASAL), then the cells were permeabilized with 10 μg/mL Digitonin, and mitochondrial respiratory rates measured after the subsequent addition of glutamate and malate (GM$_L$), ADP to stimulate respiration (GM$_P$) along with succinate to stimulate complex II (GMS$_P$), uncoupler (CCCP) to measure maximal electron transport (GMS$_E$), rotenone to block complex I (S$_E$) and Antimycin A to inhibit complex III. The residual oxygen consumption after rotenone and Antimycin A treatment was subtracted from all values shown here. OXPHOS oxidative phosphorylation, ETC electron transfer system capacity. All error bars are mean ± s.e.m. For all proteomic data (**a, c**), statistical significance was derived from two-tailed empirical Bayes moderated $t$-statistics performed in limma on total proteome, see Supplementary Data 3. All other data (**b, d, e**) were analysed by two-way ANOVA with Sidak's multiple comparisons test.

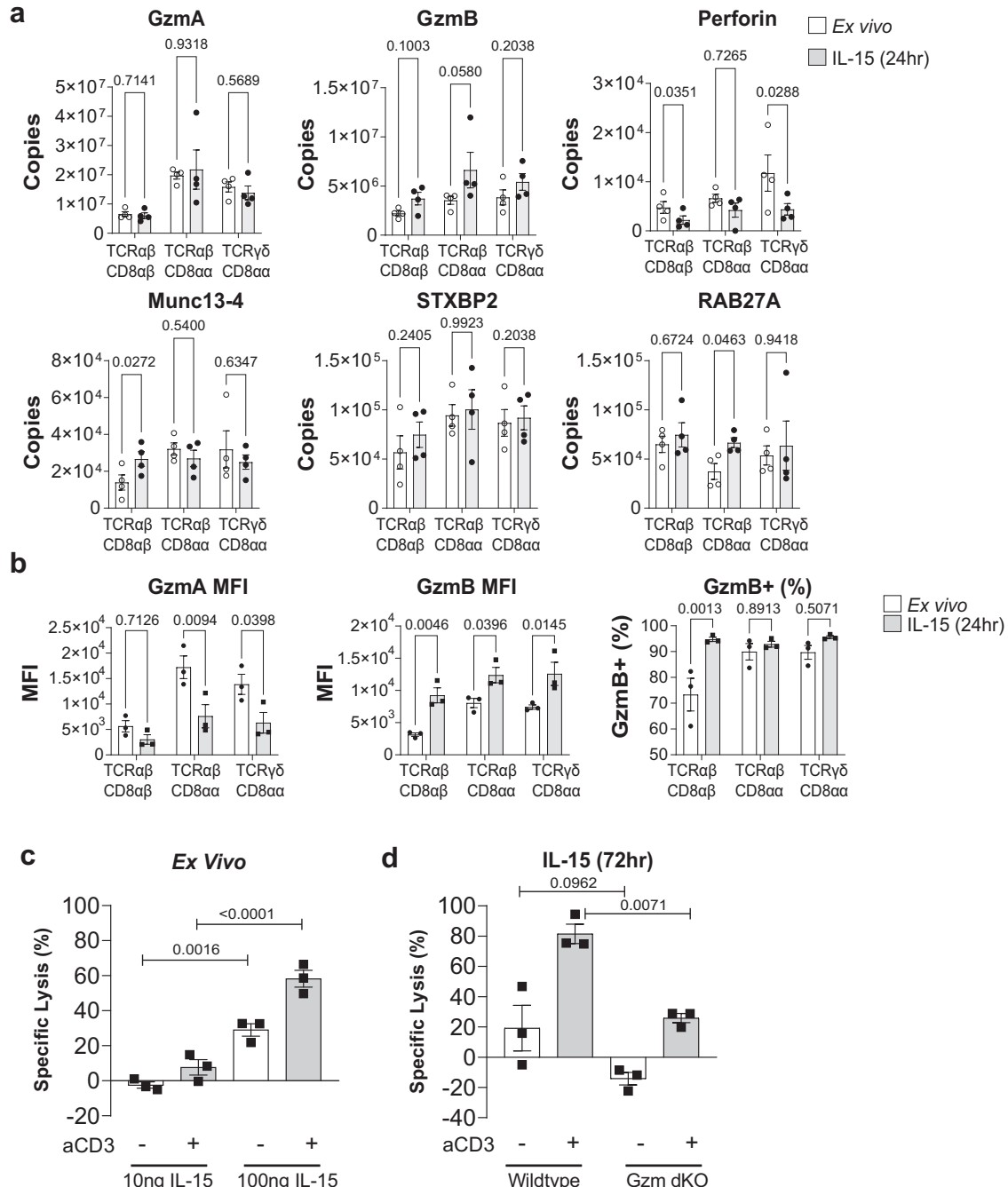

**Fig. 5 IL-15 potentiates IEL cytotoxicity. a** Estimated protein copy number/cell of cytotoxic molecules granzyme A (GzmA), granzyme B (GzmB), Perforin, Munc13-4, STXBP2 and RAB27A. Data are shown for all IEL subsets ± 24 h IL-15/Rα stimulation ($n = 4$ biologically independent samples), statistical significance was derived from two-tailed empirical Bayes moderated $t$-statistics performed in limma on total proteome, see Supplementary Data 3. **b** Flow cytometric analyses of intracellular GzmA and GzmB in freshly isolated IEL compared to those cultured with 100 ng/mL IL-15/Rα for 24 h. Data are presented as mean fluorescence intensity (MFI) and percentage positive cells for GzmB (gating strategy shown in Supplementary Fig. 6c), ($n = 3$ biologically independent experiments), analysed by two-way ANOVA with Sidak's multiple comparisons test. **c** Luciferase-transduced K562 cells were co-cultured for 24 h with freshly isolated IEL at an effector to target (E:T) ratio of 40:1. Cells were treated with either 10 ng/mL or 100 ng/mL IL-15/Rα (±aCD3). **d** Luciferase-transduced K562 cells were co-cultured for 24 h with freshly isolated IEL or WT and GzmA/B dKO IEL that had been pre-treated with 100 ng/mL IL-15/Rα (±aCD3) for 72 h, at an E:T ratio of 40:1. **c, d** Bar graphs represent the percentage of specific lysis for each condition, ($n = 3$ biologically independent experiments) data were analysed by one-way ANOVA, with Sidak's multiple comparisons test. All error bars are mean ± s.e.m.

of the TCR via anti-CD3. To investigate whether IEL require Granzymes to kill, we performed cytotoxic killing assays using $Gzma^{-/-}Gzmb^{-/-}$ (GzmA/B dKO) mice. In contrast to WT, GzmA/B dKO IEL treated with IL-15 failed to kill any of the target cells and even had the opposite effect of increasing K562

cell viability. This effect was only poorly reversed by TCR stimulation (Fig. 5d). These results suggest that GzmA/B are crucial for IEL killing and that other cytotoxicity-inducing molecules such as TRAIL and FasL do not replace granzymes in this context. Taken together, these data indicate that IL-15-

driven cytotoxicity is mediated by granzymes, and that upregulation of GzmB is unlikely to be the main mechanism involved.

**IL-15 upregulates multiple activating and inhibitory receptors on IEL.** Previous studies have reported that IEL derived from patients with CeD had elevated expression of activating NK cell receptors NKG2D and CD94[9,11]. We only observed a modest increase of NKG2D exclusively in TCRαβCD8αα IEL, and CD94 exclusively in TCRγδCD8αα IEL (Fig. 6a, b). Using flow cytometry, we confirmed that NKG2D was not expressed by IEL in the resting state, but expression was upregulated in a small percentage (~2–8%) of TCRαβCD8αα IEL specifically following IL-15/Rα treatment, and this did not increase drastically even after 72 h of culture (Fig. 6a). Similarly, CD94 was expressed mainly by ~5–15% of TCRαβCD8αα and TCRγδCD8αα IEL (Fig. 6b). Thus, CD94 and NKG2D are unlikely to be direct targets of IL-15 in the mouse small intestine.

IEL are known to express a range of other surface receptors and the proteomic data revealed that many of these were upregulated following IL-15/Rα stimulation in all IEL subsets (Fig. 6c). A number of these receptors have the potential to impact upon the activation and cytotoxic effector function of IEL such as adhesion molecules (e.g. ICAM2, Integrin α4, L1CAM) and various activating receptors, such as the junctional adhesion molecule-like (JAML) receptor and CD100 (Semaphorin 4D), both co-stimulatory receptors for TCRγδ T cells in the skin[32–35], and CD226 (DNAM-1), a cytotoxic activator of NK cells[36] (Fig. 6d). On the other hand, we also noted that IL-15/Rα led to the upregulation of various inhibitory receptors such as LAG3, LILRB4 and CD96 (Fig. 6e). Thus, the potential of IL-15 to activate IEL in vivo may be driven through increased adhesion to IEC, and the complex balance of both activating and inhibitory signals received through multiple receptors besides NKG2D and CD94.

**Identification of PIM1/2 kinases as regulators of IEL responses to elevated IL-15 signals.** Among proteins that were not expressed in IEL ex vivo but upregulated substantially by strong IL-15/Rα signals were the PIM kinases, PIM1 and PIM2 (Fig. 7a). PIM proteins are serine/threonine kinases transcriptionally regulated downstream of JAK/STAT signalling[37]. Using immunoblotting, we confirmed a clear induction of the two isoforms of PIM1, and all three isoforms of PIM2 in IEL stimulated with IL-15/Rα (Fig. 7b). Despite no detectable protein expression of PIM1 and PIM2 in ex vivo IEL, mRNA for both *Pim1* and *Pim2* were detected in ex vivo IEL, and this was significantly upregulated upon IL-15 stimulation (Supplementary Fig. 2c). While PIMs have been shown to play a key role in proliferation, cell survival and protein synthesis, their functions are highly cell type and context-specific and a role in IEL has not yet been defined[38]. To assess the contributions of PIM kinases to IL-15-stimulated IEL, we isolated IEL from $Pim1^{-/-}/Pim2^{-/Y}$ (PIM1/2 dKO) mice. Lack of PIM1 and PIM2 expression was confirmed via immunoblotting in splenic CD8 T cells, which had normal pSTAT5 levels (Supplementary Fig. 2a, b). PIM-deficient mice are healthy and have normal complement of lymphocytes in the spleen (Supplementary Fig. 2d). As IL-15 is crucial for IEL survival in vivo, it was possible that IL-15 might require PIM kinases for survival. However, IEL numbers and composition were normal in PIM1/2 dKO mice (Fig. 7c), and their survival was unperturbed in response to low levels of IL-15 (Fig. 7d). Conversely, responses to high levels of IL-15/Rα were severely impaired in PIM1/2 dKO mice. WT IEL numbers increased over 96 h in response to 100 ng/mL IL-15/Rα in vitro, whereas PIM1/2 dKO IEL failed to increase in numbers comparatively (Fig. 7e), and cell division was

completely abolished in PIM1/2 dKO mice (Fig. 7f). Strikingly, defective proliferation in PIM1/2 dKO lymphocytes appears specific to IEL, as activated splenic T cells proliferated normally in the presence of IL-2 (Supplementary Fig. 2e, f), another γc family cytokine that also induces the PIM kinases[39]. We also assessed the individual contributions of PIM1 and PIM2 and found that loss of either PIM1 or PIM2 on their own did not have as strong an effect on IEL proliferation (Supplementary Fig. 3a, b).

PIM kinases can also regulate protein synthesis in cancer cells[40]. Hence, we assessed cell growth in PIM-deficient IEL, and found that dKO IEL failed to grow in size and granularity as compared to WT IEL exposed to high levels of IL-15 (Fig. 7g). PIM regulation of growth was also specific to IEL as splenic CD8 T cell growth was unperturbed in the absence of PIMs (Supplementary Fig. 2g). As previously shown, IEL express high levels of GzmA and GzmB in the resting state, however, PIM dKO IEL had significantly lower expression of GzmB (Fig. 7h) and failed to upregulate GzmB further in response to IL-15/Rα (Fig. 7i). GzmA levels were normal in PIM dKO IEL (Supplementary Fig. 2h). The mammalian target of Rapamycin (mTOR) regulates protein synthesis and growth in T cells, and our data revealed mTORC1 activity in IL-15-stimulated IEL, as measured by phosphorylation of the mTOR target ribosomal protein S6 (RPS6), despite only a modest increase in RPS6 copy numbers (Supplementary Fig. 4a, b). Treatment of IEL with Rapamycin, an mTORC1 inhibitor, blocked IEL growth and proliferation downstream of IL-15/Rα, and marginally blocked GzmB upregulation (Supplementary Fig. 4c, d, e). Importantly, in the absence of PIM kinases, mTORC1 was not active, as measured by phosphorylation of S6 (Supplementary Fig. 4f). These data indicate that the PIM kinases may regulate growth and protein synthesis in IEL by regulating mTOR function.

Given the observed functions of PIM kinases downstream of IL-15/Rα signalling in mouse IEL, we next asked whether human IEL also upregulate PIM kinase expression in the context of IL-15 and/or CeD. Analyses of recently published RNA-Seq datasets on human TCRαβCD8αβ IEL cell lines, and on ex vivo IEL stimulated with soluble IL-15 indicated that human IEL upregulated *Pim1* and *Pim2* mRNA upon IL-15 stimulation (ref. [15] and Supplementary Fig. 5a). Published microarray analyses on T cell clones derived from human IEL and peripheral blood lymphocytes (PBL) from CeD patients revealed that both *Pim1* and *Pim2* mRNA are higher in IEL compared to PBL (ref. [12] and Supplementary Fig. 5b). To test whether increased mRNA expression of *Pim1* and *Pim2* corresponds to evidence of increased protein expression, we examined the immunohistochemical expression of PIM1 kinase by comparing duodenal biopsies in CeD versus normal controls. PIM1 staining was more intense in IEL in CeD biopsies compared to the weak staining in normal controls, a feature broadly present across the biopsies compared (Fig. 7j). Together these data indicate that upregulation of PIM kinases by IL-15 may be a conserved pathway for regulating IL-15 responses in IEL.

## Discussion

IEL are one of the least understood T lymphocyte subsets, with limited insights into their biology and regulation. Using high-resolution mass spectrometry, we reveal the IL-15-regulated proteomes of three distinct IEL subsets; TCRαβCD8αβ, TCRαβCD8αα and TCRγδCD8αα. By focussing on IL-15, a key cytokine that activates IEL function, we characterise the pathways and modules regulating IEL activation, proliferation and proteome remodelling. We observed that the major effect of IL-15 complexes on IEL is to trigger their proliferation, not only by inducing the cell cycle machinery, but also by activating all the

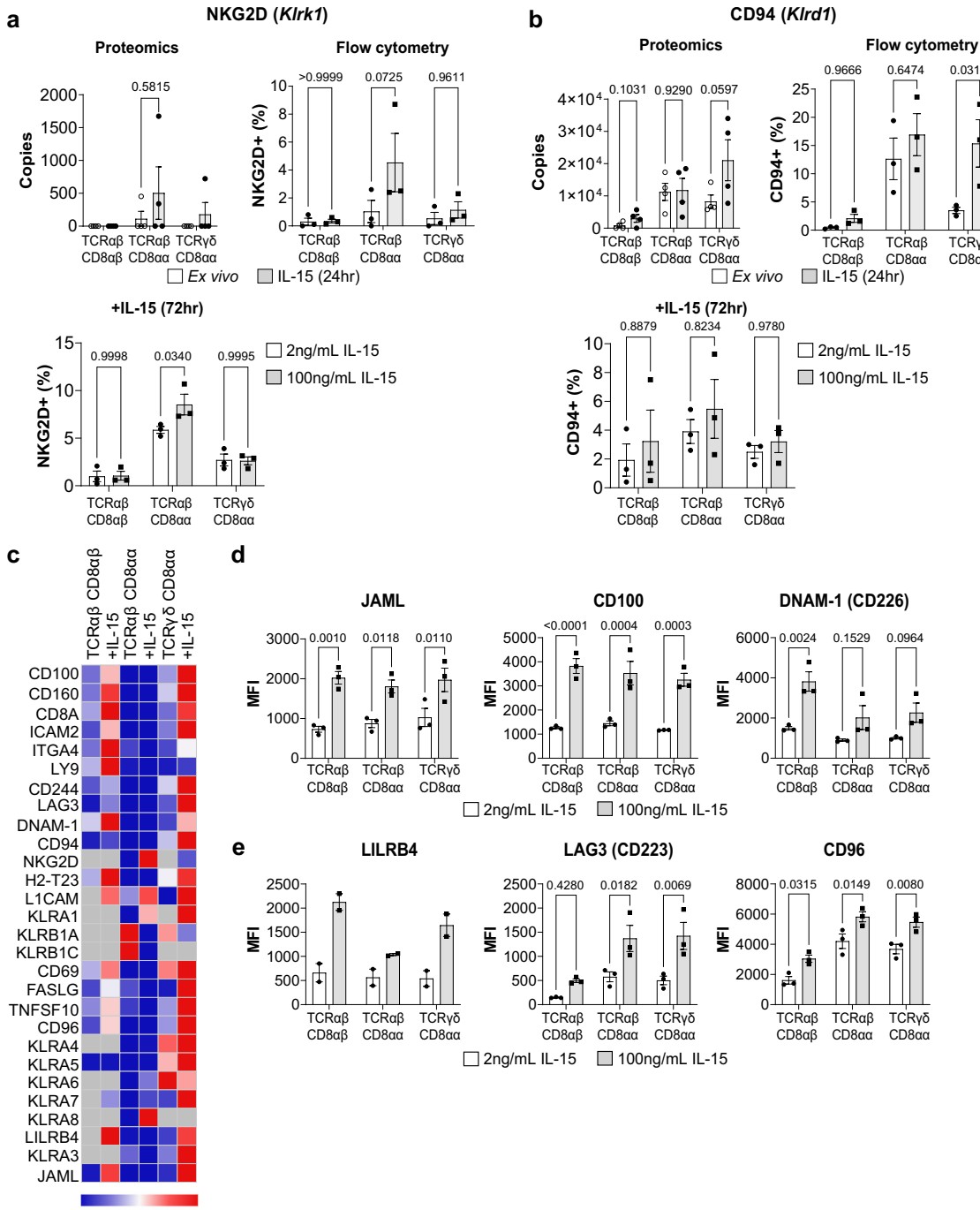

**Fig. 6 Activating and inhibitory receptor expression in response to IL-15/Rα stimulation.** Estimated protein copy number/cell of **a** NKG2D and **b** CD94, with corresponding flow cytometric analyses on either ex vivo vs 24 h IL-15-stimulated IEL (top left panels) or IEL that had been cultured in 2 ng/mL or 100 ng/mL IL-15/Rα for 72 h (bottom left panels). Flow cytometry data (*n* = 3 biologically independent experiments) are presented as percentage positive cells following gating on IEL subsets (gating strategy as in Supplementary Fig. 1a). **c** Row normalized heatmap showing the expression profile of a manually curated list of surface receptors identified in the proteomics across IEL subsets. Grey squares indicate undetectable expression. **d, e** IEL were cultured in 2 ng/mL or 100 ng/mL IL-15/Rα for 72 h and assessed for their expression of various surface receptors identified in the heatmap. Flow cytometric data are shown as mean fluorescence intensity (MFI) for **d** activating receptors; JAML, CD100 and DNAM-1, and **e** inhibitory receptors; LILRB4, LAG3 and CD96. Gating strategy as in Supplementary Fig. 1a (*n* = 3 biologically independent experiments, except for LILRB4 (*n* = 2 biologically independent experiments). All error bars are mean ± s.e.m. For all proteomic data (top right panels **a, b**), statistical significance was derived from two-tailed empirical Bayes moderated *t*-statistics performed in limma on total proteome, see Supplementary Data 3. All flow cytometry data (**a, b, d, e**) were analysed by two-way ANOVA with Sidak's multiple comparisons test.

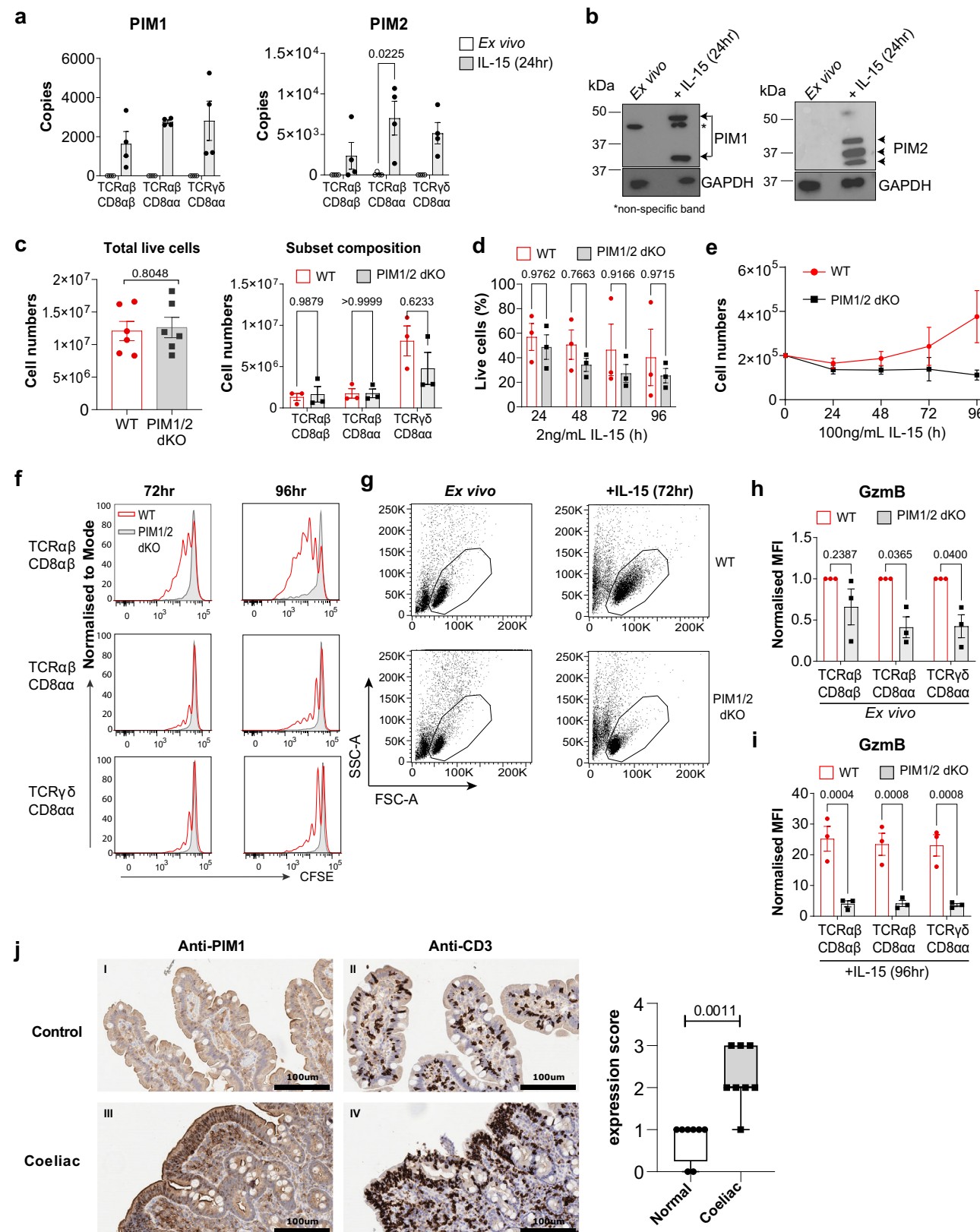

accompanying changes required, such as protein and nucleotide synthesis, ribosome biogenesis, and targeted bioenergetic activation to fuel these changes, largely regulated by the PIM kinases. IL-15 also induced cell surface proteins involved in epithelial interactions, particularly cell adhesion molecules and cytotoxic activators that regulate IEL function.

With no known endogenous antigenic stimuli, it has been difficult to define the mechanisms necessary for IEL activation. IL-15 can induce cytolytic effector function in human IEL in the context of certain target cells[7,41]. It has previously been suggested that IL-15 triggers IEL cytotoxic activity by upregulating expression of GzmB, and the activating receptor NKG2D, both in human and mouse

**Fig. 7 PIM kinases regulate IEL responses to strong IL-15/Rα stimulation. a** Estimated copy number/cell of PIM1 and PIM2 kinases in IEL ± 24 h IL-15/Rα stimulation ($n = 4$ biologically independent experiments), statistical significance was derived from two-tailed empirical Bayes moderated $t$-statistics performed in limma on total proteome, see Supplementary Data 3. **b** Immunoblot data (representative of $n = 3$ biologically independent experiments) showing PIM1 (right) and PIM2 (left) expression in ex vivo IEL or 24 h IL-15-stimulated IEL. Antibodies against GAPDH were used as a loading control. **c** Bar graphs show the absolute cell counts (left; $n = 6$ biologically independent experiments) and subset composition (right; $n = 3$ biologically independent experiments) of IEL isolated from WT (red) and PIM1$^{-/-}$/PIM2$^{-/Y}$ (PIM1/2 dKO) (grey) mice (gating strategy shown in Supplementary Fig. 6c). **d** The percentage of live IEL from either WT or PIM1/2 dKO mice that were cultured in 2 ng/mL IL-15/Rα for 96 h ($n = 3$ biologically independent experiments). Percentages were calculated from the number of live cells following IL-15/Rα treatment divided by the number of cells seeded for culture (1 million/mL) every 24 h. **e** Line graph shows the cell numbers of IEL from either WT or PIM1/2 dKO mice that were cultured in 100 ng/mL IL-15/Rα for 96 h ($n = 3$ biologically independent experiments). **f** IEL were isolated from WT and PIM1/2 dKO mice and stained with CellTrace CFSE prior to stimulation with 100 ng/mL IL-15/Rα for 4 days. Every 24 h cells were stained for subsets and CFSE expression was analysed by flow cytometry (gating strategy as in Supplementary Fig. 1a). The discrete peaks in the histograms represent successive generations of live, DAPI-negative IEL ($n = 3$ biologically independent experiments). **g** Dot plots (representative of $n = 3$ biologically independent experiments) show the forward scatter (FSC) vs side scatter (SSC) of live IEL ex vivo as compared to IEL cultured in 100 ng/mL IL-15/Rα for 72 h from both WT (top) and PIM1/2 dKO (bottom) mice. IEL that were **h** derived ex vivo and **i** cultured in 100 ng/mL IL-15/Rα from both WT and PIM1/2 dKO mice were stained for intracellular GzmB expression (gating strategy as in Supplementary Fig. 1a), presented as mean fluorescence intensity (MFI) normalized to **h** WT controls or **i** cells cultured in 2 ng/mL IL-15/Rα ($n = 3$ biologically independent experiments). **j** Representative images of normal duodenal ($n = 8$) or duodenal biopsies with histological features of coeliac disease ($n = 8$) stained with anti-PIM1 and anti-CD3 antibodies (scale bar = 100 μm). I and II (top panels) show a biopsy of normal control duodenum and III and IV (bottom panels) show features consistent with coeliac disease. Boxplot demonstrates semi-quantitative assessment of PIM1 immunohistochemical staining intensity from biopsies, whiskers are minima to maxima with all points shown. All error bars are mean ± s.e.m. For (**c**, right panel) a two-tailed unpaired $t$-test with Sidak's multiple comparisons test and **j** two-tailed Mann–Whitney test was used to derive statistical significance. (**c**, left panel), **d, h, i** were analysed by two-way ANOVA with Sidak's multiple comparisons test.

IEL[12,42]. However, we find only modest upregulation of NKG2D, on a very small proportion of IEL. We also reveal that IEL express the complete cytotoxic machinery, including GzmB, prior to IL-15 activation, so it is not clear how further IL-15-induced upregulation of GzmB would activate IEL cytotoxicity. Undeniably, there are differences between human and mouse IEL. For example, NKG2D is not constitutively expressed on IEL in mice, whereas it is on resting human IEL[11,14]. Further, TCRαβCD8αβ induced IEL, rather than natural IEL subsets, are the most predominantly expressed subset in humans[2] and the primary mediators of tissue destruction in CeD. Despite these apparent differences, we find that IL-15 also increased the cytolytic activity of mouse IEL dramatically, particularly when combined with TCR stimulation. Moreover, mice overexpressing IL-15 in the intestine display signs of villous atrophy, suggesting that IEL can be activated to kill IEC in the presence of elevated levels of IL-15[43]. Yet even in these settings, less than 10% of IEL were shown to express NKG2D[42–44]. Therefore, we suggest that other mechanisms must be contributing to IL-15 driven cytotoxicity of IEL.

Our data revealed upregulation of various activating receptors, such as JAML and CD100 in response to 24 h IL-15 stimulation. JAML was recently reported to act as a co-stimulatory receptor for epidermal TCRγδ T cells, increasing proliferation and cytokine production upon interaction with its ligand CAR on keratinocytes[33]. Similarly, CD100 was shown to specifically regulate intestinal TCRγδ IEL activation in the context of colitis and epidermal wound repair[34,35]. CD226 is also expressed on human IEL, and its activation on NK cells and CD8 T cells has been described to trigger cytotoxicity, proliferation, and cytokine production[36,45]. In our proteomic data, these receptors were not only expressed by TCRγδ IEL but shared by all IEL subsets and IL-15 stimulation commonly increased their expression, highlighting potential mechanisms for IL-15-induced activation of IEL. Another possible explanation for the increased cytotoxic activity of IL-15 stimulated IEL is that IL-15 lowers the signalling threshold required for TCR or other co-receptor activation.

One fundamental insight from our proteomic data was the IL-15-induced upregulation of PIM1 and PIM2. Despite the low-level IL-15 signalling that occurs in vivo, we did not detect PIM1 or PIM2 expression in ex vivo IEL, suggesting that there is a threshold of IL-15 signalling required to trigger the expression of these kinases. Using PIM1/2-deficient mice we have shown that

IEL depend on both PIM kinases for their proliferative expansion and growth in response to IL-15 in vitro. Interestingly, PIM1/2-deficient IEL appeared phenotypically normal in unchallenged mice, except for a reduction in GzmB expression. It is also noteworthy that in the absence of PIM kinases, IL-15 could not induce upregulation of GzmB. These data suggest that the PIM kinases also control the cytotoxic activation of IEL.

Intraepithelial lymphocytosis and villous atrophy, both hallmarks of CeD, are thought to be driven by IL-15. Currently the only treatment available for CeD patients is a life-long gluten-free diet. However, issues due to cost, non-compliance with the diet, and gluten contamination cause symptoms in up to 30–50% of CeD patients. Additionally, in a small percentage of CeD patients, even being gluten-free does not reduce inflammation, leading to RCeD[46]. Hence, there is an urgent need for treatments for CeD, RCeD and associated lymphomas. Currently, IL-15 blocking antibodies are being investigated as a treatment strategy[47], with limited success[48,49], thus identifying new targets for small molecule inhibitors would be beneficial. Our finding that PIM1 is overexpressed in CeD biopsies suggest that PIM kinase inhibitors may be effective for treatment of CeD and RCeD. PIM kinase inhibitors are currently in clinical trials for the treatment of certain types of haematological malignancies and solid cancers[50,51], thus there is clear potential for these drugs to also be used for the treatment of CeD and its complications.

This study provides a comprehensive proteomic analyses of the effects of strong IL-15 stimulation on IEL. The effects of IL-15 on IEL receptor expression and biosynthetic pathways, among others, are unstudied pathways that warrant further investigation. In particular, the identification of the PIM kinases as key regulators of IEL biology is a pivotal finding of this study. Further research will reveal the mechanisms by which PIM kinases orchestrate IEL biology and immune responses, and whether this a feature specific to IEL. The proteomics data and insights obtained in this study will serve as an important resource, not only for the study of IEL in CeD, but also in the context of infection, inflammatory bowel diseases and cancer, where IL-15 is overexpressed.

## Methods

**Mice.** All mice were bred and maintained with approval by the University of Dundee ethical review committee, under a UK Home Office project license

(PD4D8EFEF) in compliance with U.K. Home Office Animals (Scientific Procedures) Act 1986 guidelines. C57BL/6 J male mice were purchased from Charles Rivers and acclimatised for a minimum of 10 days prior to use in experiments at 50–60 days old. Pim1$^{-/-}$/Pim2$^{-/Y}$ (PIM1/2 dKO), Pim1$^{-/-}$ (PIM1 sKO) and Pim2$^{-/Y}$ (PIM2 sKO) mice[38] on C57BL/6 J background were obtained from D. Cantrell, University of Dundee, Scotland with permission from A. Berns, Netherlands Cancer Institute (NKI-AVL), Amsterdam. PIM1/2 dKO experiments were both male and female, aged 70–300 days old with age and sex-matched non-littermate wild-type controls. PIM1 sKO proliferation experiments were both male and female mice aged 70–300 days old with age and sex-matched littermate wild-type controls. PIM2 sKO proliferation experiments were male mice aged 70–80 days old with age and sex-matched littermate wild-type controls. Gzma$^{-/-}$/Gzmb$^{-/-}$ (GzmA/B dKO) mice on C57BL/6 J background were obtained from J. Pardo, University of Zaragoza, Spain[52]. GzmA/B dKO killing assay were both male and female mice aged 80–110 days old, with co-housed age and sex-matched C57BL/6 J wild-type controls.

Mice were maintained in a standard barrier facility on a 12 h light/dark cycle at 21 °C, 45–65% relative humidity. Mice were tested negative for all pathogens on the current FELASA list. Mice were maintained in individually ventilated cages with corn cob or Eco Pure chips 6 and sizzler-nest material and fed an R&M3 diet (Special Diet Services, UK) and filtered water ad libitum. Cages were changed at least every two weeks.

**Isolation of IEL from the intestinal epithelium.** IEL were isolated as described[53]. Briefly, small intestines were dissected from proximal duodenum to terminal ileum and using a gavage needle, flushed with 20 mL cold PBS to remove luminal contents. Intestines were cut longitudinally and then transversely into small 5–10 mm pieces. The pieces were placed into 25 mL complete RPMI media (RPMI + 10% FBS, 1% Pen/Strep & L-glutamine) with 1 mM DL-Dithiothreitol (DTT) and incubated on a shaker for 40 min at RT. After centrifugation and vortexing in complete RPMI media the cells were passed through a 100 μm strainer and isolated cells were centrifuged in a 36%/67% Percoll/PBS density gradient at 700 × g for 30 min with no brake. IEL appear as diffuse layer at the interface between the two Percoll concentrations.

**Sample preparation for mass spectrometry.** For proteomics experiments, four biological replicates were generated and IEL were isolated from eight mice per biological replicate (male, C57BL/6, aged 10-12 weeks). IEL were isolated from the small intestine as previously described. For further enrichment of the CD8α+ IEL population, following Percoll density gradient centrifugation, an EasySep Release PE positive selection kit (STEMCELL Technologies, #17656) was used with a PE-conjugated anti-mouse CD8α antibody (BioLegend, #100708) as per the manufacturer's instructions. Cells were stained with PerCP eFluor 710 (TCRγδ), APC (TCRβ), PE (CD8α), FITC (CD8β) and PE-Cy7 (CD4) for isolation of pure populations of TCRγδCD8αα, TCRαβCD8αα and TCRαβCD8αβ IEL using fluorescent activated cell sorting (FACS) (Supplementary Fig. 1A, full list of antibodies provided in Supplementary Data 5). We purified ~10million TCRγδCD8αα, ~5 million TCRαβCD8αα and ~5 million TCRαβCD8αβ per biological replicate. Two million cells from each purified cell population were pelleted and snap frozen in liquid N$_2$ and the remaining cells treated with 100 ng/mL IL-15/Rα for 24hrs as described below then similarly pelleted and snap frozen.

IEL cell pellets (both IL-15-treated and untreated) were lysed in 200 μL lysis buffer (4% SDS, 10 mM TCEP, 50 mM TEAB (pH 8.5)). Lysates were boiled and sonicated (15 cycles of 30 s on/30 s off) and protein concentrations determined by EZQ Protein Quantitation Kit (Invitrogen, #R33200). Lysates were then alkylated with iodoacetamide (IAA) for 1 h at room temperature in the dark. For protein clean-up, 200 μg SP3 beads[54] were added to lysates before elution in digest buffer (0.1% SDS, 50 mM TEAB (pH 8.5), 1 mM CaCl$_2$) and digested with LysC and Trypsin, each at a 1:100 (μg enzyme:protein) ratio. Peptide clean-up was performed according to the SP3 protocol. Samples were resuspended in 2% DMSO and 5% formic acid. For fractionation, off-line high pH (9.5) reverse phase chromatography was used. Using an Ultimate 3000 HPLC (Thermo Scientific), peptides were separated into 16 concatenated fractions, then manually pooled to 8 orthogonal fractions, dried and eluted in 5% formic acid for analysis by LC-MS. Samples were sent to MRC-PPU Mass Spectrometry facility, University of Dundee, where each fraction was analysed by label-free quantification (LFQ) using an LTQ-Orbitrap Velos Pro mass spectrometer running Xcalibur software (v. 2.1.0SP1.1160) (Thermo Scientific) with a 240-minute gradient per fraction.

**Processing and statistical analysis of proteomics data.** The MS raw data files were processed with MaxQuant version 1.6.8.0 as described[39], against mouse reviewed proteome from Swissprot with isoforms, downloaded in Aug 2019. Minimum peptide length was set to 6, and proteins were quantified on unique + razor peptides with the following modifications included: Oxidation (M); Acetyl (Protein N-term); Deamidation (NQ). The dataset was filtered to remove proteins categorised as "contaminants", "reverse" and "only identified by site" (a summary detailing protein identification in each sample is provided in Supplementary Data 4). Pearson's correlation coefficients for each of the samples was calculated using the associated protein intensities and subsequent heatmaps were plotted in

the 'proteus' package[55] in RStudio. MaxQuant-derived data were converted into estimated copy numbers per cell as described[19]. Protein (i) copy number (CN$_{(i)}$) was calculated using the following formula: N$_A$*5.5*MS-intensity$_{(i)}$/ (total histone MS intensities*MW$_{(i)}$, where MW is the protein molecular weight in Daltons, N$_A$ is Avogadro's Constant, and 5.5 pg is the estimated weight of 2n DNA content in mouse cells. The protein content was calculated using the following formula: 5.5*total MS intensities/total histone MS intensities = protein mass (pg/cell). For differential expression analyses, p-values and fold changes were calculated on log$_2$-normalized copy numbers using the limma package in R[56,57]. All comparisons were made using the default conditions for the package. P-value adjustment was performed using the Benjamini-Hochberg (BH) method, and associated q values were calculated using the Bioconductor package 'qvalue', a list of which can be found in Supplementary Data 3. Elements with p-values < 0.05 were considered significant. Fold-change thresholds were determined using 1 standard deviation of the mean log2[fold change] in each IEL subset as a guide. Complete analysed data are in Supplementary Data 3. Following DE analysis, only proteins with >2 peptides quantified were retained to reduce the impact of false positive identifications. For functional annotation analysis of proteins identified, the Database for Annotation, Visualization and Integrated Discovery (DAVID) Bioinformatics resources 6.8[58,59] was used (https://david.ncifcrf.gov/tools.jsp). Heatmaps were generated using the Morpheus tool (https://software.broadinstitute.org/morpheus) from the Broad Institute.

**Cell culture of IEL and cytotoxicity assay.** Isolated IEL were further enriched using an EasySep Mouse CD8α positive selection kit II (STEMCELL technologies, #18953) as per the manufacturer's instructions. IEL positively enriched for CD8α expression were resuspended in culture medium (RPMI + 10% FBS + 1% Pen/Strep, L-glutamine, sodium pyruvate, non-essential amino acids, 2.5% HEPES and 0.1 mM β-mercaptoethanol) with various concentrations of Mouse IL-15/IL-15R Complex Recombinant Protein (referred to as IL-15/Rα) (ThermoFisher, #14-8152-80). These cells were seeded in a round-bottom 96-well plate at 1 million cells/mL (2 × 10$^5$ cells per well) and incubated at 37 °C, 10% CO$_2$. For rapamycin treatment, 20 nM of InSolution Rapamycin (Merck, #553211) was added to cells in the presence of 100 ng/mL IL-15/Rα and cultured for specified amount of time in figures. For analysis of proliferation, CellTrace CFSE Cell Proliferation Kit (Invitrogen, #C34554) was used; briefly, cells were treated with 5 μM CFSE at 37 °C for 10 min before quenching in ice cold PBS + 15% FCS and put into culture as described above. For the bioluminescence-based cytotoxicity assay, luciferase-expressing K562 cells (provided by Dr. S. Minguet, Freiburg) were plated at a concentration of 5 × 10$^3$ cells per well in a 96-well flat bottom plate in triplicates. 75 μg/mL D-firefly luciferin potassium salt (Biosynth, #L-8220) was added to the K562 cells and bioluminescence was measured using a PHERAstar plate reader to establish the bioluminescence baseline. For a maximal cell death (positive control), triton X-100 was added to K562 cells at a final concentration of 1%. For spontaneous death (negative control), culture medium was added to K562 cells. For the test, WT or GzmA/B dKO IEL that had been cultured for 72hrs in 100 ng/mL IL-15/Rα were added to K562 cells at a 40:1 effector-to-target (E:T) ratio and incubated for 24hrs at 37 °C, 10% CO$_2$. Bioluminescence was measured as relative light units (RLU). Percentage specific lysis was calculated with the following formula: % specific lysis = 100 × (average spontaneous death RLU − test RLU)/(average spontaneous death RLU − average maximal death RLU).

**Cell culture and treatments for splenocytes and lymph nodes.** Single-cell suspensions from WT and PIM1/2 dKO spleens were activated with purified anti-CD3 + anti-CD28 [both 0.5 μg/mL, clones 2C11 and 37.51, respectively, (eBioscience, #100302/BioLegend, #102116)] in the presence of recombinant human IL-2 [20 ng/mL (Proleukin, Novartis)]. After 48 h cells were washed out of activation media, then split daily into fresh media and IL-2 (20 ng/mL) to a density of 0.3 million cells/mL. CD8 T cell numbers of WT and PIM1/2 dKO T cells were counted on a FACSVerse daily from day 2 of the culture onwards. Lymph node single-cell suspensions from WT and PIM1/2 dKO were labelled with 5 μM Cell-Trace Violet Cell Proliferation Kit (CTV) (Invitrogen, #C34557) for 20 min at 37 °C, washed then activated with anti-CD3 + anti-CD28 [both 0.5 μg/mL clones 2C11 and 37.51, respectively, (eBioscience/Thermo Fisher Scientific)] at 100,000 cells per well in 96-well flat-bottomed plate in 200 μL total volume for 4 days. Cells were stained with Propidium Iodide (PI) (Sigma, #P4170) at a final concentration of 0.2 μg/mL before surface staining with CD4 and CD8 as described in 'flow cytometry' section. Cell proliferation was assessed daily by flow cytometry.

**Measurement of extracellular metabolites in cell culture.** IEL were isolated and enriched for CD8α as previously described before plating at 200,000 cells/well in a round-bottom 96-well plate in 200 μL medium (glucose-free RPMI to which 2 mM glucose, 2 mM L-glutamine 10% dialysed FBS, 1% Pen/Strep, sodium pyruvate, non-essential amino acids, 2.5% HEPES and 0.1 mM β-mercaptoethanol were added). Cells were cultured at 37 °C, 10% CO$_2$ and every 24 h 10 μL of medium was removed, diluted in 190 μL of PBS and further diluted 2.5x for glucose measurements. Samples were frozen and stored at −20 °C. At the end of the experiment, samples were thawed and 50 μL aliquots transferred to a 96-well assay plate (Corning,#3903). Each sample was plated in duplicate for each metabolite. The

metabolites were then detected using the Lactate-Glo and Glucose-Glo assays (Promega, #J5021, #J6021) as per manufacturer's instructions. Luminescence was recorded using a PHERAstar plate reader and measured as relative light units (RLU).

**Flow cytometry**. Cells were plated at $2 \times 10^5$ cells per well to a 96-well dish for staining. FC block (BioLegend, #147605) was added to each well for 5 min before cells were incubated with monoclonal antibodies (mAb) against cell surface markers for 15 min in the dark at 4 °C. Cells were stained with the following antibodies specific for murine: CD45 [clone 30.F11 (BioLegend)], TCRβ [clone H57-597 (BioLegend)], TCRγδ [clone GL3 (BioLegend or eBioscience)], CD4 [clone RM4-5 (BioLegend)], CD8α [clone 53-6.7 (BioLegend)], CD8β [clone H35-17.2 (eBioscience)], CD122 (IL-15Rβ) [clone TM-b1 (eBioscience)], NKG2D [clone CX5 (BioLegend)], CD94 [clone 18d3 (BioLegend)], JAML [clone HL4E10 (BioLegend)], CD100 [clone BMA-12 (BioLegend)], CD223 (LAG3) [clone eBioC9B7W (eBioscience)] and CD85k (LILRB4) [clone H1.1 (BioLegend)], CD226 (DNAM-1) [clone 10E5 (eBioscience)], CD69 [clone H1.2F3 (eBioscience)], CD96 [clone 3.3 (BioLegend)], CD71 [clone RI7217 (BioLegend)], CD98 [clone RL388 (BioLegend)]. DAPI (Invitrogen, #D1306) was used as cell viability dye for live cells. Fixed cells were treated with LIVE/DEAD Fixable Blue Dead Cell Stain Kit (Invitrogen, #L34962) for dead cell discrimination. For fixation before intracellular staining, cells were treated with 2% PFA at 37 °C for 10 min. For intracellular staining, cells were washed in 1X permeabilization buffer (eBioscience, #00-8333-56) before 1 h incubation at room temperature with mAbs specific for mouse: GzmB [clone GB12 (eBioscience)] or GzmA [clone GzA-3G8.5 (eBioscience)]. For detection of phospho-STAT5 and phospho-S6 ribosomal protein by flow cytometry, cells were fixed with 2% PFA prior to any surface stains, permeabilised with 90% ice cold methanol and incubated with Rabbit monoclonal phospho-STAT5 (Tyr694) [clone C11C5 (Cell Signaling Technology)] (1:200 dilution), or Rabbit monoclonal Phospho-S6 Ribosomal Protein (Ser235/236) [clone D57.2.2E (Cell Signaling Technology)](1:25 dilution) for 30 min at room temperature. Cells were then incubated with an anti-rabbit DyLight 649-conjugated donkey secondary Ab [clone Poly4064 (BioLegend)] for 30 minutes at room temperature (1:500 dilution). To measure DNA synthesis or protein synthesis, cells were treated with 10 μM base-click 5-Ethynyl-deoxyuridine (5-EdU) (Sigma, #BCK-FC647-50) for 2 h, or 20 μM O-propargyl-puromycin (OPP) (JenaBioscience, #NU-931-05) for 15 minutes, respectively. For OPP assays, a negative control was pre-treated with 0.1 mg/mL Cycloheximide solution (Sigma, #C4859) for 30 min before adding OPP. Cells were then harvested (~1 million cells per condition), fixed with 4% PFA and permeabilised with 0.5% triton X-100 before undergoing a copper catalysed click chemistry reaction with Alexa 647-azide using the EdU Flow Cytometry Kit 647 (Sigma, #BCK-FC647-50). Cells were then stained with surface markers as described above and resuspended in FACS buffer (PBS + 1% FBS (+15 μg/mL DAPI for cell cycle analysis)) and analysed by flow cytometry to determine the degree of incorporation of EdU or OPP. All data were acquired on a FACSVerse flow cytometer with FACSuite software (version 1.0.5.3841, BD Biosciences) or a FACS LSR Fortessa flow cytometer with DIVA software (version 8.0.1, BD Biosciences). Data were analysed using FlowJo software (TreeStar). A list detailing all antibodies and dilutions can be found in Supplementary Data 5.

**Fluorescent cell barcoding (FCB)**. IEL were isolated and enriched for CD8+ as described above. Cells were resuspended at a concentration of 1 million cells/mL and 500 μL plated in a 24 flat bottom well plate. Cells were warmed at 37 °C for 30 min before stimulated with different concentrations of IL-15/Rα for 3 h at 37 °C. After stimulation cells were directly fixed with 500 μL PFA 4% 10 min at 37 °C before permeabilisation with 90% ice cold methanol. During methanol permeabilisation, each sample was stained with a mix of various concentrations of amine-reactive fluorescent dyes for 40 min, on ice before quenching with PBS + 0.5%BSA (v:v). Pacific Blue Succinimidyl Ester (Invitrogen, #P10163) was used at a concentration of 0 μg/ml, 11.1 μg/ml or 100 μg/ml and DyLight800 NHS Ester (Thermo Scientific, #46421) at a concentration of 0 μg/ml or 25 μg/ml. Barcoded samples were then pooled and stained for intracellular pSTAT5 and surface markers as described in the "flow cytometry" section. Data were acquired using CytoFlex flow cytometer (Beckman Coulter) with CytExpert (v2.4.0.28) software and analysed using FlowJo Software. Data were analysed using the "forward deconvolution method"[60]. Briefly, samples were differentiated based on the fluorescence intensities of each dye and then individual samples were analysed for pSTAT5 expression.

**High-resolution respirometry**. Mitochondrial respiration was measured in digitonin-permeabilised IELs that enables studying mitochondria in their architectural environment. CD8+ IEL were isolated and cultured as described above, in 10 ng/mL or 100 ng/mL IL-15/Rα for 44 h. The analysis was performed in a thermostatic oxygraphic chamber at 37 °C with continuous stirring (Oxygraph-2k, Oroboros instruments, Innsbruck, Austria). Approximately 1 million cells were placed in Mir05 respiration medium (0.5 mM EGTA, 3 mM MgCl₂, 60 mM lactobionic acid, 20 mM taurine, 10 mM KH₂PO₄, 20 mM HEPES, 110 mM D-Sucrose and 1 g/L of fatty acid free Bovine Serum Albumin [BSA]; pH = 7.1) in the oxygraphic chamber. Respiration protocol was adapted from the Substrate-uncoupler-

inhibitor titration protocol number 11 (SUIT-011)[61]. Briefly, after the determination of the BASAL oxygen consumption in absence of any substrate, cells were permeabilized with digitonin (10 μg/10⁶ cells). Substrates and inhibitors were then added sequentially to determine respiratory rates. LEAK respiration was first measured by adding glutamate (10 mM) and malate (2 mM) to the chamber (GM$_L$). OXPHOS respiration was then determined by first adding 2.5 mM ADP (GM$_P$). The OXPHOS GM$_P$ state records electron flow from the type N-pathway (NADH-generating substrates to complex I) to Q-junction that feeds electrons to complexes III and IV. The maximal OXPHOS respiration rate GMS$_P$ was then measured by adding 25 mM succinate, with both type N-pathway to Q and type S-pathway (succinate, substrate of complex II) to Q being stimulated. Electron transfer system capacity (ETC) was then assayed by adding incremental doses of Carbonyl cyanide m-chlorophenyl hydrazine (CCCP) (0.05 μM steps, GMS$_E$). Complex I was blocked with 0.5 μM rotenone to determine the Succinate-pathway control state (S$_E$). Finally, residual oxygen consumption (ROX) was determined in presence of 2.5 μM Antimycin A. ROX was subtracted from oxygen flux as a baseline for all respiratory states. Respiratory rates were expressed as pmol O₂ x s⁻¹ x million cells⁻¹.

**Immunoblots**. IEL were isolated and ex vivo samples were cultured with 10 ng/mL IL-15/Rα for 2 h to allow for a basal level of PIM expression before harvesting 5 million cells and pelleting. Stimulated IEL were cultured as described above with 100 ng/mL IL-15/Rα for 24 h and then similarly refreshed in medium + IL-15/Rα for 2 h before harvesting. Cell pellets were lysed at between 50 and 60 million cells/ mL RIPA lysis buffer (50 mM Tris pH 7.4, 1% (v/v) NP-40, 0.5% (w/v) Na deoxycholate, 0.1% (w/v) SDS, 150 mM NaCl, 2 mM EDTA, 50 mM NaF, 1 mM TCEP, 5 mM Na-pyrophosphate, 10 mM Na-β-phosphoglycerate, 1 mM Na Orthovanadate, cOmplete mini EDTA-free protease inhibitor tablet (Roche)) for 10 min on ice before centrifuging for 12 min at 4 °C. Protein concentrations were determined for IEL lysates using the Coomassie (Bradford) Protein Assay Kit (Thermo Scientific, # 23200) as directed. Lysates were boiled with NuPAGE LDS sample buffer (4X) for 3 min at 100 °C. 20 μg of IEL lysate and ~140,000 CTL lysate per well were separated using a 4% stacking/10% resolving SDS-PAGE gel at 120 V for 100 min. Following separation, proteins were transferred to a PVDF membrane for 90 min at 300 mA. Membranes were then blocked in 5% (w/v) milk (Marvel)/ PBS-Tween for 30 min at RT. Following blocking, membranes were washed and incubated with primary monoclonal antibodies; anti-mouse PIM1 [clone 12H8 (Santa Cruz Biotechnology)] or anti-mouse PIM2 [clone 1D12 (Santa Cruz Biotechnology)] both used at 1:100 in 5% (w/v) milk (Marvel)/PBS-Tween. Monoclonal anti-rabbit GAPDH [clone 14C10 (Cell Signalling Technology)] was used as a loading control (1:1000 in 3% BSA/PBS-Tween). PanSTAT5, rabbit polyclonal antibody [clone D2O6Y (Cell Signalling Technology)] and phospho-STAT5 (Tyr694), rabbit polyclonal antibody (Cell Signalling Technology, cat # 9351) both used at 1:1000 in 3% BSA/PBS-Tween. For PIM1 and PIM2 antibody specificity and selectivity (Supplementary Fig. 2a, b), gels were cut down to centre to blot for phospho-STAT5 and PIMs from halves of the same membrane. Phospho-STAT5 membrane was then stripped with stripping buffer (200 mM glycine, 3.5 mM SDS, 1% Tween20) for 30 min, washed with PBS-Tween and reblotted for panSTAT5. Membranes were washed out of primary and incubated with HRP-conjugated secondary anti-mouse or anti-rabbit IgG (Cell Signalling Technology) at 1:5000 in 5% (w/v) milk (Marvel)/PBS-Tween, or 1:2000 in 5% BSA/PBS-Tween, respectively. Membranes were exposed to Clarity Western ECL substrate or Clarity Max Western ECL Substrate (BioRad, #1705061, #1705062) and signals were detected using a ChemiDoc imaging system. A list detailing all antibodies and dilutions can be found in Supplementary Data 5.

**RNA extraction and qPCR**. RNA was isolated using the PureLink RNA mini kit (Invitrogen, # 12183018 A) as per manufacturer's instructions, and complementary DNA (cDNA) was generated using PrimeScript RT Reagent Kit with gDNA Eraser (Takara, #RR047B). Real-time PCR was performed with the TB Green Premix Ex Taq II (Takara, #RR820W) on a Biorad CFX384 Touch Real-Time PCR machine. CD3ε was used as a reference gene. Data were analysed using the ΔCt method. Primers were as follows: PIM1 5′- CAAACTGTCTCTTCAGAGTG -3′ (forward) and 5′-GTTCCGGATTTCTTCAAAGG-3′ (reverse), PIM2 5′-ATGTTGACCAAG CCTCTG-3′ (forward) and 5′-CGGGAGATTACTTTGATGG-3′ (reverse), CD3ε 5′- GAGAGCAGTCTGACAGATAGGAG-3′ (forward) and 5′GAGGCAGGAGA GCAAGGTTC-3′ (reverse). Details of primers used are provided in Supplementary Table 1.

**Human tissue and immunohistochemistry**. All human tissues were obtained with written informed consent from donors under the governance of and with ethical approval from the NHS Research Scotland Biorepository in Tayside. Formalin-fixed paraffin embedded (FFPE) duodenal biopsies were obtained from patients who had undergone oesophageal-gastro duodenoscopy in which the histological features were subsequently in keeping with coeliac disease ($n = 8$) compared to controls in which features were subsequently within normal limits ($n = 8$). Biopsies selected for coeliac disease staining were restricted to those with intraepithelial lymphocytosis, sub-total to total villous atrophy in the correct clinical (and serological) context for coeliac disease. Normal biopsies were those showing no

histopathological abnormality. Sections from FFPE blocks (nominally 4 microns thick) were cut onto superfrost plus slides (VWR International Ltd) and dried for 1 h at 60 °C. Antigen retrieval and de-paraffinization was performed using DAKO EnVision FLEX Target Retrieval solution (high pH) buffer (Agilent Technologies, #K8004) in a DAKO PT Link. Sections were blocked in Flex Peroxidase-Blocking Reagent (Agilent Technologies, #SM801) and incubated overnight at 4 °C with anti-PIM1 [clone 12H8 (Santa Cruz)] at a dilution of 1:25 and anti-CD3 (A0452, Agilent) at a dilution of 1:100. Immunostaining using DAKO EnVision FLEX system (Agilent Technologies) on a DAKO Autostainer Link48 was carried out according to manufacturer's protocol. DAKO substrate working solution was used as a chromogenic agent for 2 × 5 min and sections were counterstained with EnVision FLEX haematoxylin. Sections from mantle cell lymphoma known to stain positively were included in each batch and negative controls were prepared by replacing the primary antibody with DAKO antibody diluent. Slides were scanned to produce whole slide images using the Leica Aperio CS2 system, and representative image snapshots were then captured from the whole slide images at equivalent to ×200 magnification. PIM1 stained coeliac and normal control slides were randomised and the intensity of staining of PIM1 on the original slides were scored by light microscopic assessment using a Nikon Eclipse Ni microscope. A semi-quantitative scoring system from 0 to 3+ was used to evaluate the average staining of intraepithelial lymphocytes across the whole of each biopsy. A list detailing all antibodies and dilutions can be found in Supplementary Data 5.

**Statistical analyses**. Differential expression analyses from the proteomic data were performed as outlined in the 'Processing and statistical analysis of proteomics data' section. Further analysis on data derived from the proteomic data and all validation experiments were analysed by two-way ANOVA, with Sidak's multiple comparison test unless otherwise stated in the figure legends. Statistical analyses were carried out using GraphPad Prism v.8. For bar graphs, symbols on bars represent independent biological replicates. Results are shown as mean ± s.e.m. $P$-values <0.05 were considered to denote significance.

**Reporting summary**. Further information on research design is available in the Nature Research Reporting Summary linked to this article.

## Data availability
All analysed proteomics data generated during this study are provided in Supplementary Data 3, with further analyses in Supplementary Data 1, 2 and 4. The mass spectrometry raw data have been deposited in the ProteomeXchange Consortium via the PRIDE partner repository[62] with the dataset identifier PXD025891. Gene expression data from human samples were obtained from publicly available Gene Expression Omnibus datasets GSE120904 and GSE4592. A source data file is provided with this manuscript for all figures, excluding data pertaining to; Fig. 1c which is provided in full in supplementary Data 3; data pertaining to Fig. 1d and e are provided in supplementary Data 1; data pertaining to Fig. 1f is provided in supplementary Data 2; and data pertaining to Supplementary Fig. 1c and d are provided in supplementary Data 3. All other data supporting this study can be obtained from the corresponding author upon request. Source data are provided with this paper.

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

## Acknowledgements

This study was supported by a Wellcome Trust PhD studentship to O.J.J. (215309/Z/19/Z) and by the Wellcome Trust and Royal Society (Sir Henry Dale Fellowship to MS, 206246/Z/17/Z). J.M.M. is supported by EMBO (ALTF 1543-2015), and Marie Skłodowska-Curie grant agreement (705984). We are very grateful to D. Cantrell, Dundee, for providing us access to the PIM1/2 dKO mice, and to D. Cantrell, N. Cerf-Bensussan, Y. Kulathu for many useful discussions. We would like to thank A. Rennie, A. Gardner and R. Clarke for cell sorting and technical assistance, D. Campbell, R. Gourlay, J. Varghese, R. Nirujogi for mass spectrometric analyses in the MRC-PPU, the biological resource unit, and G. Gilmour and E. Emslie for genotyping. We gratefully acknowledge the support of Dr. Sharon King and the Tayside Tissue Biorepository.

## Author contributions

M.S. conceptualized and designed the study, O.J.J. and M.S. curated and analysed proteomic data, O.J.J., M.V., J.M.M., F.S. and M.S. performed experiments and analysed data. S.E.B. performed IHC on human biopsies, and J.W. evaluated and scored kinase expression in the biopsies, A.G.L. provided data validation. O.J.J. and M.S. prepared figures and wrote the manuscript with input from all co-authors.

## Competing interests

The authors declare no competing interests.
