## [Peer Review File · Nature Communications]

In their review of the first version of this manuscript, reviewer #5 uploaded a Microsoft Excel file with their review. As this had multiple sheets it could not be properly formatted in to this pdf file so has been redacted.

REVIEWER COMMENTS

Reviewer #5 (Remarks to the Author):

Because the authors did not respond clearly to my second comment I decided to check the available data and using these to recalculate the total protein per cell and the copy number values. Using the data from Table S3 I got values of total protein content per cell that are clearly deviating and generally higher in comparison to the values shown in the panel B of Fig.1. (see attached Table Calculations). For example, for TCRabCD8aa+IL15 samples the values are 182, 204, 237, 85 [pg/cell] (or ug/1,000,000cells) whereas in the Fig 1b the values range between ~70 and ~170. To follow this, I have checked the content of histones and I found that the intensity values (Table S3) are not useful for applying the proteome (histone) ruler because only one of the core histones the H4 was regularly identified with comparable intensities across the samples. The other 3 core histones (2a, 2b, and 3) were almost irregularly observed. Compared to histone H4, in most samples they ms signals are very low or zero. In contrast, the content of Histone H1 variants was very high. I would like to recall that in non-dividing/slowly growing cells the expected mass ratio of histones H1/H2a/H2b/H3/H4 is always roughly 1:1:1:1:1. For more details, see attached Excel spreadsheet with my calculations.

Based on such data the usage of the histone ruler does not make any sense. The reported values for protein per cells as well the copy numbers are incorrect. I do not believe that the performed LC/MS analysis did not enabled measurement of abundances of histones. Most probably improper data processing (MQ, Perseus) has led to this. I cannot check further this issue because authors did not provide a link to their raw ms data.

The missing values for histones raise the question on comprehensiveness of the data.

Jacek R. Wisniewski

RESPONSE TO REVIEWER COMMENTS

Reviewer #5 (Remarks to the Author):

Because the authors did not respond clearly to my second comment, I decided to check the available data and using these to recalculate the total protein per cell and the copy number values. Using the data from Table S3 I got values of total protein content per cell that are clearly deviating and generally higher in comparison to the values shown in the panel B of Fig.1. (see attached Table Calculations). For example, for TCRabCD8aa+IL15 samples the values are 182, 204, 237, 85 [pg/cell] (or ug/1,000,000cells) whereas in the Fig 1b the values range between ~70 and ~170. To follow this, I have checked the content of histones and I found that the intensity values (Table S3) are not useful for applying the proteome (histone) ruler because only one of the core histones the H4 was regularly identified with comparable intensities across the samples. The other 3 core histones (2a, 2b, and 3) were almost irregularly observed. Compared to histone H4, in most samples they ms signals are very low or zero. In contrast, the content of Histone H1 variants was very high. I would like to recall that in non-dividing/slowly growing cells the expected mass ratio of histones H1/H2a/H2b/H3/H4 is always roughly 1:1:1:1:1. For more details, see attached Excel spreadsheet with my calculations.

Based on such data the usage of the histone ruler does not make any sense. The reported values for protein per cells as well the copy numbers are incorrect. I do not believe that the performed LC/MS analysis did not enabled measurement of abundances of histones. Most probably improper data processing (MQ, Perseus) has led to this. I cannot check further this issue because authors did not provide a link to their raw ms data.

The missing values for histones raise the question on comprehensiveness of the data.

We are very grateful to Prof. Wisniewski for going through the mass spec quantification data in detail and noticing that there was an error in our analyses. After going through the analysis process, we have realised that the mistake we had made was to quantify the proteins based on unique peptides only, and not on unique + razor peptides. We have now rerun the data through MaxQuant to correct this mistake. We have updated the protein content and copy numbers according to the method advised in *Wiśniewski, J.R. (2017). Label-Free and Standard-Free Absolute Quantitative Proteomics Using the "Total Protein" and "Proteomic Ruler" Approaches. Methods Enzymol 585, 49–60.* We have included an Excel sheet with these calculations as an attachment to this response, and detailed the calculations in the Methods section of the manuscript.

With the new data we find that the histones were identified and quantified in all samples, and the protein content was approximately 40-60pg/cell, which is more in line with similar sized lymphocytes in other studies and in the original proteomic ruler paper (*Wiśniewski, J.R., Hein, M.Y., Cox, J., and Mann, M. (2014). A "proteomic ruler" for protein copy number and concentration estimation without spike-in standards. Mol Cell Proteomics 13, 3497–3506*). In parallel, we ran the samples through the histone ruler plugin on Perseus, and obtained similar results, while also confirming that the selection of histones used for normalisation were accurate. We also calculated the ratios between histones and have included this data in the Excel sheet. We do not find the ratios between H1, H2a, H2b, H3 and H4 to be within 1:1:1:1, as they are quite variable. However, we respectfully disagree with the reviewer that the ratios of the histone intensities should be 1:1:1:1. Since each histone has different unique peptides and we don't know how well they fly and get fragmented in the mass spectrometer, and whether they are all quantified, it would be difficult to get this ratio to be the same for all samples. Further, there are many isoforms and variants of histones expressed, a large number of posttranslational modifications (PTMs) on histones, which can also affect their relative quantification. I believe that is why the histone ruler method uses the summed total histone intensities as a group rather than relying on 1 specific histone. In fact, in the original paper (*Wiśniewski, et al 2014*), the ratio

of histones in the 4 cell lines, A549, HepG2, PC-3 and U87-MG are also not found at 1:1:1:1 ratio. I have provided this data as comparison on the Excel sheet. Overall, we find the total histone content of the cells to be quite consistent, between 9-15% of the total protein content. Based on this data, and the improved quantification, we feel that the use of the histone ruler method is justified here.

Importantly, the new copy numbers and quantifications have resulted in improved coverage (>7500 proteins), reduced variability, and therefore we now identify a substantially higher proportion of proteins as differentially expressed after IL-15/R treatment. For proteins upregulated >2-fold, this number increased from 164 to 244. Most of the proteins identified with the previous analyses were also identified in these, along with several new ones. Notably, rerunning these DE proteins through functional annotation clustering revealed the same pathways as previously identified to be regulated by IL-15. Therefore, we are happy to note that all of the findings and conclusions from our previous version remain unchanged. We have redone all the figures with the new copy numbers and analyses as detailed below. We hope that the reviewer will find that the reanalysed data suitable for publication.

I also include here the reviewer login details to access the raw MS data on the Proteome Xchange server. I apologise for not including it in the previous response.

Dataset identifier: PXD019711

Username: reviewer60245@ebi.ac.uk

Password: yvwKv9gJ

Changes to figures

Fig 1b – Protein content updated

Fig 1c. volcano plots updated due to re-analysis of data/new copy numbers

Fig1d&e – Venn diagrams changed

Fig1f – new functional annotation clustering

Fig2b – new copy numbers

Fig2d – new copy numbers

Fig2e – figure changed to a schematic of the cell cycle regulator proteins involved with G1/S transition and their corresponding log₂ fold changes, to better represent significantly changed proteins

Fig 3b - Figure changed to a schematic of the 90S pre-ribosome components from the enriched KEGG pathway 'ribosome biogenesis in eukaryotes' and their corresponding log₂ fold changes

Fig3e – Updated log₂ changes of nutrient transporters

Fig 3f – new copy numbers

Fig 4a&c – new copy numbers

Fig5a – new copy numbers

Fig 6a&b- new copy numbers

Fig 6c – updated expression of receptors heatmap with new average copy numbers

Fig 7a – new copy numbers

Supplementary

FigS1c&d – new correlation of intensities and copy number heatmaps

FigS1e – new copy numbers

FigS4a – new copy numbers

REVIEWERS' COMMENTS

Reviewer #5 (Remarks to the Author):

Authors have responded to my criticism and recalculated their data.
I have no further comments.